

# Iceberg Detection Based on the Swin Transformer Algorithm and SAR Imagery: Case Studies off Prydz Bay and the Ross Sea, Antarctic

Fangru Mu[1, 2, 3], Chengfei Jiang[4], Bin Cheng[5], Keguang Wang[6], Caixin Wang[6], Yuhan Chen[3, 2], Zhiyuan Shao[7, 2] and Jiechen Zhao[7, 8, 2]*

[1] Hainan Aerospace Technology Innovation Center, Wenchang, 571300, China
[2] Laboratory for Regional Oceanography and Numerical Modeling, Qingdao Marine Science and Technology Center, Qingdao, 266400, China
[3] Qingdao Innovation and Development Base & College of Underwater Acoustic Engineering of Harbin Engineering University, Qingdao, 266400, China
[4] National Satellite Ocean Application Service, Beijing, 100081, China
[5] Finnish Meteorological Institute, Helsinki, 00101, Finland
[6] Department of Research and Development, Norwegian Meteorological Institute, Oslo, 0313, Norway
[7] First Institute of Oceanography, Ministry of Natural Resources, Qingdao, 266400, China
[8] UN Decade Collaborative Centre on Ocean-Climate Nexus and Coordination Amongst Decade Implementing Partners,
Qingdao, 266400, China

*Correspondence to*: Jiechen Zhao (zhaojiechen@outlook.com)

**Abstract.** Icebergs pose persistent hazards to maritime navigation and offshore operations. In Antarctica, grounded offshore icebergs may gradually melt, altering the local ocean stratification conditions. This in turn influences coastal ocean circulation, sea ice dynamics, and thermodynamics. Accurately identifying the spatiotemporal distribution of icebergs is essential for both

maritime operations and oceanographic research. In this study, we developed an iceberg detection algorithm based on the Swin transformer model (IDAS-Transformer). The IDAS-Transformer, along with a support vector machine (SVM) and a residual network (ResNet18), was applied to four synthetic aperture radar (SAR) images acquired over Prydz Bay and the Ross Sea, which represented a landfast ice zone, a drift ice zone, and an open ocean. The coverage area of each image was 80 km × 80 km. Manual interpretation was employed to generate reference data for algorithmic evaluation purposes. The iceberg

concentration, defined as the area occupied by icebergs per grid unit, along with the total number of icebergs and their average size, was introduced to provide a quantitative iceberg detection assessment. We found that the IDAS-Transformer performed well across various sea ice conditions, and a total of more than 800 icebergs were detected. Both the F1 scores and the kappa coefficients of the model exceeded 85%. The total number of identified icebergs and their area presented mean biases of +4.13% and +3.65%, respectively. The IDAS-Transformer outperformed the other two tested algorithms. The sea ice concentration

affects the iceberg detection process, with the main challenge being the separation of icebergs from similarly textured pack ice in complex ice-covered regions. Furthermore, distinguishing icebergs that are smaller than 160 m × 160 m among large ice floes remains difficult.

**Keywords.** Antarctica, Prydz Bay, Ross Sea, Iceberg detection, Deep Learning, Swin Transformer, Synthetic Aperture Radar imagery





## 1 Introduction

Polar regions are among the most vulnerable areas on Earth to global climate change. Ice sheets are melting at an accelerated rate (Rignot et al., 2011; Pattyn et al., 2018). The risk of ice shelf collapse and the subsequent formation of icebergs is increasing (Liu et al., 2015; Golledge et al., 2019). Approximately 130,000 icebergs exceeding 10 metres in length have been observed during the Antarctic summer over the past three decades (Orheim et al., 2023). Drifting icebergs pose significant

hazards to ship navigation.

The calving of icebergs in Antarctica discharges massive amounts of freshwater into the Southern Ocean (Silva et al., 2006; Depoorter et al., 2013; Tournadre et al., 2015). The freshwater input may exhibit considerable spatial and temporal variability, with particularly pronounced localized impacts. Fresh meltwater released from drifting icebergs can influence ocean processes and circulation patterns at local and regional scales by altering the vertical and horizontal salinity gradients, ocean stratification

effect, and depth of the mixing layer (Jongma et al., 2009; Moon et al., 2018; Cenedese et al., 2023). These changes may in turn further influence the sea surface temperature (SST) and sea ice cover area (Luckman et al., 2010; Wesche & Dierking, 2012; Merino et al., 2016). Grounded small icebergs may serve as nucleation sites that promote the development of extensive landfast sea ice (Massom et al., 2003). Model studies suggest that iceberg melt may increase the Antarctic sea ice cover area by 5–10% (Jongma et al., 2009; Merino et al., 2016), potentially exerting limited negative feedback on the global climate by

enhancing the surface albedo (Jongma et al., 2009). Melting icebergs release nutrients, increasing the primary productivity level and potentially impacting marine ecosystems (Smith et al., 2013). Hence, understanding iceberg drift, carving, and ablation processes is essential for better modelling ice–ocean interactions and marine ecosystems (Enderlin et al., 2018; Moon et al., 2018; Barbat et al., 2019b). To quantify the impact of icebergs, their spatiotemporal distribution must first be identified. Shipborne radar is an essential tool for detecting obstacles during navigation. However, distinguishing icebergs from ship

radars in ice cover regions may be difficult, and the monitoring range is limited to a small spatial scale (Barbat et al., 2021). The dense fog created by the release of moisture and the strong westerly wind zone (Gajananda et al., 2007) may further restrict the ability to observe icebergs along shipping routes. Unmanned aerial vehicles (UAVs) are good complements to ship radar for monitoring icebergs, but their operations are highly restricted by weather conditions (Briggs et al., 2020).

Satellite remote sensing is a powerful tool for monitoring icebergs. Spaceborne scatterometers can detect large icebergs in sea

ice cover areas (Stuart et al., 2011). Optical satellite instruments offer high-resolution images that can be used to identify icebergs. However, the presence of cloud cover and polar nights limits the availability of optical satellite images for iceberg detection. Additionally, distinguishing ice from snow may be difficult (Mazur et al., 2017; Braakmann-Folgmann et al., 2023). An active microwave instrument, i.e., synthetic aperture radar (SAR), provides high-spatial-resolution imagery under all sky conditions. The corresponding probability distribution function (PDF) can be generated on the basis of in situ iceberg

observations and SAR imagery acquired from the target area. Thus, SAR data have been widely used for iceberg classification (Wesche and Dierking, 2012; Asiyabi et al., 2023). However, sea ice deformation may introduce challenges when performing radar altimetry-based iceberg detection (Tournadre et al., 2008, 2012).



Icebergs are typically classified through statistical thresholding (Young et al., 1998), machine learning (ML) (Barbat et al., 2019a), and deep learning (DL) (Braakmann-Folgmann et al., 2023). Icebergs typically appear brighter than the surrounding
sea ice or open water in SAR imagery because of the double-bounce effect caused by their sharp edges and rough surfaces (Karvonen et al., 2021). A variety of threshold segmentation methods, including the latest constant false-alarm rate (CFAR)-based thresholding technique, have been developed and applied for detecting icebergs (Willis et al., 1996; Gill et al., 2001; Wesche and Dierking, 2012; Frost et al., 2016; Mazur et al., 2017). Thresholding-based classification is often limited by the presence of poor backscatter gradients due to complex sea ice or high-wind conditions (Young et al., 1998; Wesche & Dierking,
2015). ML can better address the categorized features of SAR data and has been successfully employed to detect medium and large icebergs (Barbat et al. 2019a; Koo et al. 2023). Xiao et al. (2020) compared multiple ML algorithms, along with different feature combinations and feature standardization methods in iceberg detection tasks. DL methods based on advanced neural networks address large datasets, have shown excellent target detection performance (Ren et al., 2016; Lang et al., 2025), and usually outperform ML methods (LeCun et al., 2015). DL has been used to detect icebergs (Krishnan et al., 2022; Braakmann-
Folgmann et al., 2023) and to discriminate between icebergs and ships (Yang et al., 2020). The scarcity of high-quality and large-scale labelled datasets is a bottleneck for DL.

The transformer model and its variants have become new alternatives for completing SAR-related target detection tasks (Wu et al. 2023). A Swin transformer-based neural network (GLA-STDeepLab) was developed for detecting ice shelf calving fronts (Zhu et al., 2023). It is capable of identifying precise dynamic information about glaciers and ice shelf fronts. The calving front
at an ocean boundary presents similar conditions and challenges to those encountered by icebergs (Braakmann-Folgmann et al., 2023). The Swin transformer can better cope with the unique characteristics of SAR data, such as their high spatial resolutions, speckle noise, and complex textures, which require models that are capable of extracting multiscale features while effectively mitigating noise. The Swin transformer can capture both local and global features through hierarchical feature extraction and a shifted window-based self-attention mechanism, so it is well suited for detecting iceberg with SAR data.

On the basis of the successful applications of the Swin transformer neural network, we developed a novel iceberg detection algorithm called "IDAS-Transformer", which was adapted from the Swin transformer. We applied several architectural modifications to the Swin transformer to further accommodate the characteristics of Antarctic SAR imagery, with the aim of increasing its detection accuracy and noise resilience. These changes included tailored adjustments to mitigate speckle noise and optimizations of the depth and channel configurations of the model to improve its detection performance. To our
knowledge, the Swin transformer has not been applied to iceberg detection tasks.

In this study, the IDAS-Transformer, along with a support vector machine (SVM) and a residual network (ResNet18), was applied to SAR images acquired over Prydz Bay and the Ross Sea, covering landfast ice regions, drift ice areas, and the open ocean. The objectives of this work were (1) to develop a new IDAS-Transformer iceberg detection system; (2) to identify icebergs with complex surface backgrounds; and (3) to quantitatively assess the iceberg distribution in the target domain and
compare the obtained results with those of other established iceberg detection methods (an SVM and ResNet18). We demonstrate that the IDAS-Transformer can offer quality iceberg detection service for marine operations.



## 2 Data and methods

### 2.1 Investigated domain and datasets

This study was focused on Prydz Bay and the Ross Sea as the primary research areas, where the Chinese Antarctic scientific

stations (Zhongshan Station and Qinling Station) are located. Zhongshan Station was established in February 1989 at 69°22′S, 76°22′E in the Larsemann Hills on the coast of Prydz Bay. Qinling Station, China's newest Antarctic research station, was inaugurated in January 2024 and is situated at 74°56′S, 163°42′E on Inexpressible Island in the Ross Sea. In addition to being essential navigational routes for China's Xuelong research vessels during Antarctic expeditions, Prydz Bay and the Ross Sea represent key regions of interest for global scientific studies concerning ocean–ice shelf interactions.

In this study, we employed SAR images derived from the Sentinel satellite constellation series of the Copernicus observation program of European Earth. Specifically, four Sentinel-1 Level-1 ground range-detected (GRD) products acquired in the extra-wide swath (EW) mode with HV polarization (horizontal transmit and vertical receive) were selected. The EW mode provides a 400-km-wide swath with a 20 m (range) × 40 m (azimuth) native resolution, enabling both high-resolution feature extraction and wide-area surveillance. An analysis conducted during data preprocessing revealed that HV polarization offers better

contrast and backscatter differences between iceberg and non-iceberg features (sea ice), especially for smaller icebergs, than does HH polarization (horizontal transmit and horizontal receive) (Fig. 1). This finding was validated through a cross-comparison conducted on 20 randomly selected SAR images, with over 90% of the results confirming the superior performance of HV. Consequently, HV-polarized Sentinel-1 SAR imagery was chosen as the primary data source for this study.

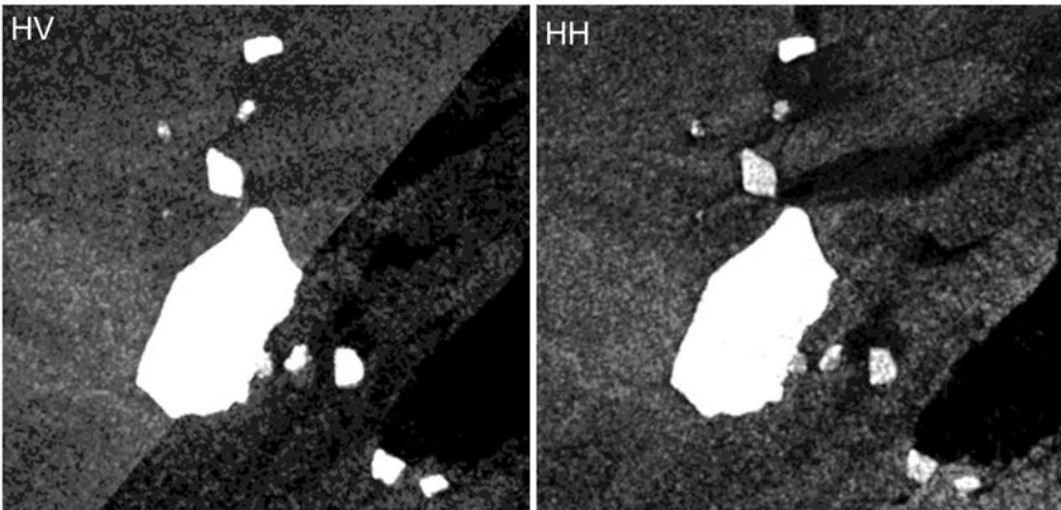

**Figure 1: Iceberg differences between Sentinel-1 SAR imagery preprocessed under HV (left) and HH (right) polarizations.**

To ensure clarity, an 80 km × 80 km area of interest (AOI) was extracted from each image of Prydz Bay and the Ross Sea for a detailed analysis, as highlighted by the enlarged red zone in Fig. 2. The detailed information is presented in Table 1. The selection criteria for AOIs were aimed at capturing a gradient of sea ice conditions—from landfast ice (SIC = 75%) to high-



concentration ice floes (SIC = 73%), low-concentration ice floes (SIC = 21%), and open water (SIC = 0%)—while also
encompassing different seasons (in March, September, and December). This selection strategy ensured spatial and temporal
representativeness while providing a robust evaluation environment for model validation purposes.

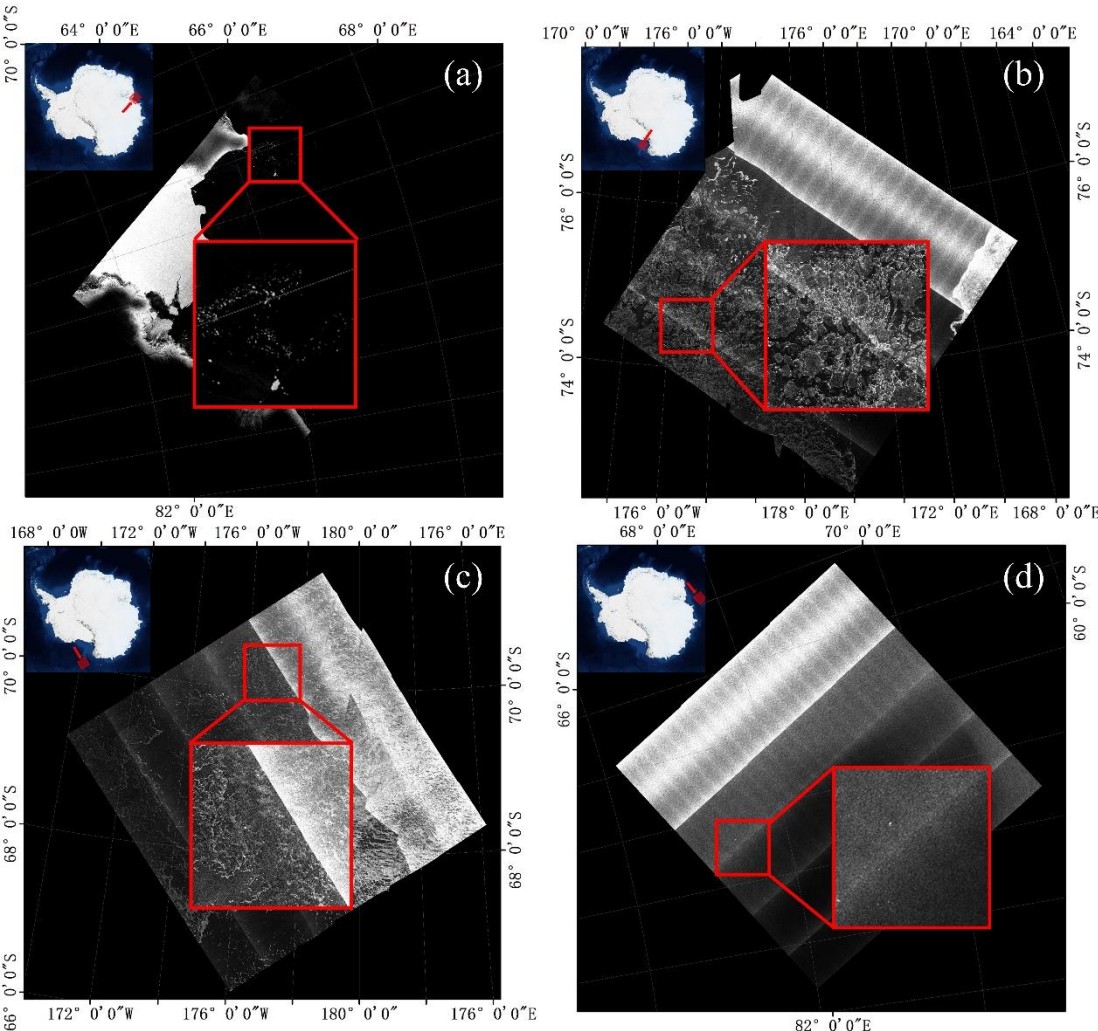

**Figure 2: The zoomed red areas represent AOIs 1–4 in (a–d), respectively. The images for AOI1–4 were captured on September 30, 2023, December 27, 2023, December 27, 2023, and March 26, 2023, respectively. The location of each scene (red arrow) is indicated**
**in the overview map located in the top-left corner of each panel. Image credit for the overview maps: NASA Worldview (https://worldview.earthdata.nasa.gov/).**





**Table 1. Detailed information about the four AOIs.**

| AOI | Latitude | Longitude | Time | Region | SIC |
|-----|----------|-----------|------|--------|-----|
| 1 | 67.03°S–67.91°S | 69.14°E–71.44°E | September 30, 2023 | Prydz Bay | 75% |
| 2 | 74.08°S–74.82°S | 175.51°W–178.27°W | December 27, 2023 | Ross Sea | 73% |
| 3 | 69.81°S–70.54°S | 175.83°W–177.99°W | December 27, 2023 | Ross Sea | 21% |
| 4 | 64.15°S–64.97°S | 76.09°E–77.93°E | March 26, 2023 | Prydz Bay | 0% |

Moderate-Resolution Imaging Spectroradiometer (MODIS) images were obtained from https://worldview.earthdata.nasa.gov/ with a resolution of 250 metres. They were used to determine that AOI1 was covered by landfast sea ice.

The daily averaged sea ice concentration (SIC) product derived from the Advanced Microwave Scanning Radiometer 2 (AMSR2) is operationally provided by the University of Bremen, using the 89 GHz channel. These data are generated using the ARTIST sea ice algorithm (ASI) and have a spatial resolution of 6.25 km (Spreen et al., 2008). This product is widely applied in polar ship navigation and operational sea ice forecasting. In this work, AMSR2 was used to calculate the average SICs in the four areas of interest. The average SICs were 75%, 73%, 21%, and 0%, respectively, corresponding to a landfast
ice area, a floe ice area with a high ice concentration, a floe ice area with a low ice concentration, and an ice-free area.
SAR imagery of Prydz Bay and the Ross Sea region contains various elements, such as icebergs, seawater, and sea ice. In this study, we analysed a total of 50 Sentinel-1 SAR scenes with HV polarization. These scenes were specifically categorized into two groups: icebergs and non-icebergs (background). Open water, ice floes, broken ice, and landfast ice were labelled as non-iceberg features. Regions with high brightness levels in the SAR images were labelled icebergs. To construct the dataset, we
extracted a total of 30,000 image patches from the SAR scenes, each with dimensions of 4×4 pixels (corresponding to an area of 160 m × 160 m). Instead of using a single-channel input, we replicated the same 4×4 image patch across three channels. This design choice was based on experimental results indicating that improved noise resilience and detection performance could be achieved with multichannel representations. Each patch was annotated with a label of 1 for icebergs or a label of 0 for non-icebergs. The dataset was then divided into training and validation sets, with 80% of the data (24,000 samples)
allocated for model training and 20% (6,000 samples) allocated for validation. The training set consisted of all 12000 icebergs and 12000 non-icebergs, selected randomly. The testing set contained 6000 samples. The dataset was compiled from Antarctic scenes with diverse ice conditions, including landfast ice, ice floes, and open water. Samples of icebergs and non-icebergs derived from the test dataset are presented in Fig. 3.



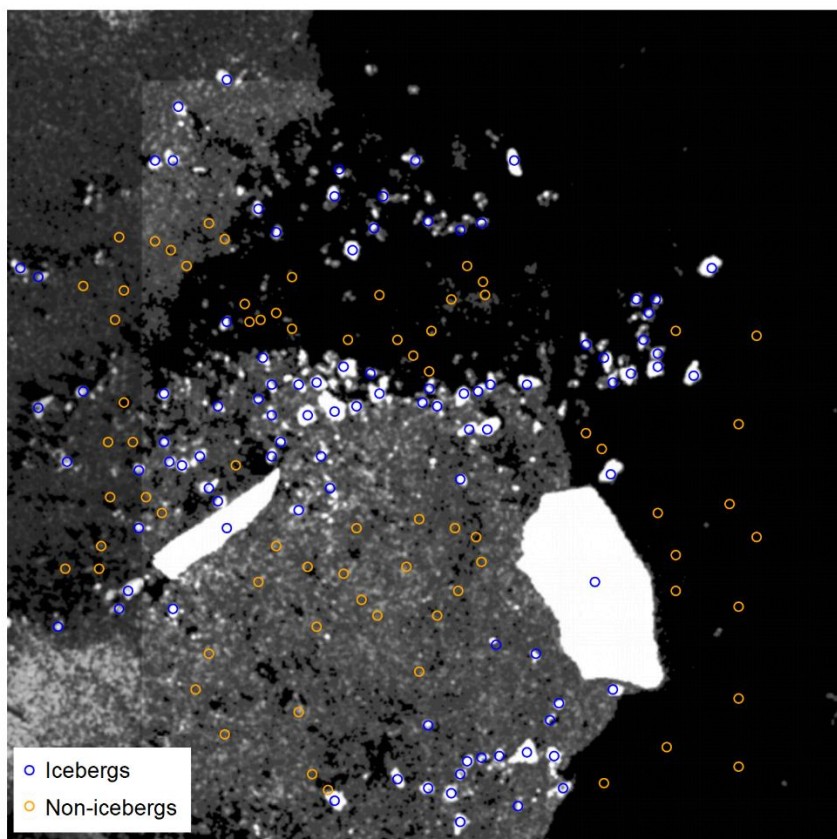

**Figure 3: Labelled icebergs and non-icebergs acquired from the Prydz Bay region and used for model training and validation purposes.**

## 2.2 Methodology

### 2.2.1 Data preprocessing

Data preprocessing was carried out using the Sentinel Application platform (SNAP) software provided by the European Space Agency (ESA). Previous studies have often included extensive preprocessing steps, such as speckle filtering and multilooking (Mazur et al., 2017; Braakmann-Folgmann et al., 2023). However, through experimentation, we found that satisfactory results could be achieved without these operations. We employed only ellipsoid correction on the SNAP to increase the accuracy and precision of the coordinate data. The initial step involved performing ellipsoid correction to project the input image into the stereographic south pole coordinate system. The Sentinel-1 (EW mode) imagery used here contained noticeable noise on the left side of the strip. To address various levels of noise interference, a linear stretching process was subsequently applied to each image. The presence of surrounding broken ice with similar characteristics, as shown in Fig. 4(a), posed challenges in terms of distinguishing between icebergs and broken ice. The linear stretching process mitigated noise interference and enhanced the degree of differentiation between iceberg features and the background, as shown in Figs. 4a–b. Finally, OpenCV





was applied for brightness adjustment purposes (linear transformation), as shown by the difference between Figs. 4b–c, to
accentuate the iceberg features. This processing procedure was manually calibrated on the basis of environmental conditions
derived from the SAR imagery, ensuring that the input imagery of the model was optimized for classification.

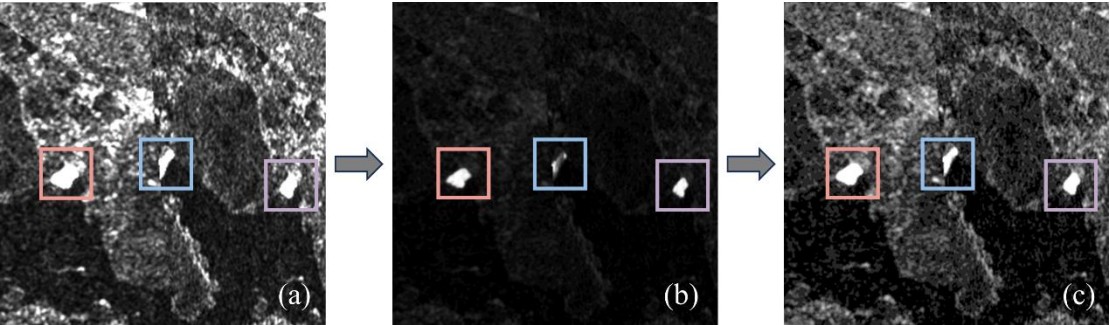

**Figure 4: An example of SAR imagery preprocessing. (a) An image produced after performing ellipsoid correction. (b) An image produced after performing linear stretching. (c) An image produced after performing linear transformation via OpenCV.**

### 2.2.2 Detection algorithm


The transformer model, which is a DL architecture based on an attention mechanism, was initially proposed by Google in 2017. A distinguishing feature of a transformer is its self-attention mechanism, which calculates attention scores for each element within a sequence. The sequence is subsequently weighted and averaged based on these scores to generate a new representation. Challenges are encountered when adapting the transformer model from language to visual tasks because of
disparities between these two domains. To address these disparities, Liu et al. (2021) proposed a hierarchical vision transformer with shifted windows, called the Swin transformer, which results in greater efficiency by limiting self-attention computations to non-overlapping local windows while also allowing for cross-window connections. These characteristics of the Swin transformer make it suitable for a wide range of vision tasks, including image classification. For more detailed information on the Swin transformer, the authors recommend the work of Liu et al. (2021).

The IDAS-Transformer model, which is an adaptation of the Swin transformer architecture, was specifically designed for detecting icebergs in Antarctic SAR imagery. This model consists of two main modules, as depicted in Fig. 5: an IDAS-Transformer architecture (Fig. 5c) and two successive Swin transformer blocks (Fig. 5d). The subsequent sections provide detailed descriptions of these modules.

The original SAR images are denoted as $x \in H \times W \times C$, where H and W signify the spatial height and width, respectively, and
C indicates the number of spectral bands. The preprocessed SAR images are divided into non-overlapping 4×4 pixel patches, the sizes of which are determined through an evaluation. Each patch, with dimensions of 4×4×3, is treated as a token, and its features are represented as a sequence of vectors. Unlike the original Swin transformer, our model omits the positional encoding process. Instead, a token embedding layer directly transforms each patch into its corresponding token. Throughout this process, the image resolution is preserved (with a downsampling factor of 1), which is essential for maintaining fine-
grained spatial details and is particularly relevant for detecting small icebergs.



The IDAS-Transformer follows a typical hierarchical architecture with four stacked stages. Within each stage, feature information is extracted through the iterative stacking of token embedding and Swin transformer blocks. The number of layers contained in each respective stage is [2, 2, 6, 2]. Given the unique characteristics of the input patch window, each layer accumulates discriminative features, emphasizing the information along the channel and spatial dimensions. This architecture

is designed to progressively increase the number of output channels from C to 8C, allowing the model to extract features with increasing complexity. The number of channels involved in each stage is represented as [32, 64, 128, 256]. These hierarchical feature maps enable the Swin transformer model to effectively capture both local and global features in SAR images, which is essential for distinguishing icebergs from the surrounding sea ice and water.

Each Swin transformer block consists of a multi-head self-attention module—using either a regular windowing configuration

(W-MSA) or a shifted windowing configuration (SW-MSA)—as well as a multi-layer perceptron (MLP). A layer normalization (LN) precedes the attention and MLP modules, and residual connections follow each component, which improves the gradient flow and stability of the model—both of which are crucial for efficiently processing SAR imagery. Two successive blocks—one with W-MSA and the other with SW-MSA—form a basic unit.

After traversing the Swin transformer blocks, the feature maps are subjected to adaptive average pooling, flattened, and finally

passed through a fully connected layer (FCL) to yield binary classification results for iceberg detection.

To evaluate the performance of the IDAS-Transformer in iceberg detection tasks, we conducted comparative experiments with an SVM and ResNet18. SVMs, which were first introduced in the 1990s for classification and nonlinear function estimation, seek to determine an optimal hyperplane in a high-dimensional feature space that maximizes the separation margin between different classes (Cortes et al., 1995). Owing to its strong generalization ability and robustness, SVMs have been widely applied

in sea ice type discrimination and ice–water classification scenarios. Given these advantages, an SVM was selected as a benchmark binary classification method for iceberg detection. In our dataset, each patch consisted of a 4×4 pixel grid across three channels, resulting in dimensions of 4×4×3. The SVM processed these inputs by extracting the relevant features and, if necessary, applying kernel functions to map the data to a higher-dimensional space, enabling the model to handle cases where linear separability could not be achieved in the original space. During training, the SVM optimized an objective function that

balanced margin maximization and regularization, yielding a decision boundary and support vectors. Once trained, the model classified new 4×4×3 images by evaluating their positions relative to the hyperplane. This enabled the SVM to effectively distinguish between iceberg and non-iceberg samples, as illustrated in Fig. 5a.

ResNet is a deep residual network proposed by Microsoft Research Asia that is broadly applied to image classification and target detection tasks. Its key idea is to introduce residual connections, which solve the problems of vanishing gradients and

exploding gradients via shortcut connections that skip one or more layers during the training processes of deep neural networks (He et al., 2016). In this study, we used ResNet18, a deep convolutional neural network, to classify iceberg and non-iceberg samples. ResNet18, with its 18 convolutional layers, employs residual connections (skip connections) to overcome the vanishing gradient problem that is often encountered in deeper networks, facilitating a more stable and efficient training procedure. The input images were 4×4×3 in size, with the initial feature extraction step performed by a 3×3 convolutional



layer that produced 64 channels; this was followed by 2×2 max pooling to reduce the spatial resolution. The network then progressed through four residual stages with 64, 128, 256, and 512 channels. Each stage included two 3×3 convolutional layers connected by skip paths, either through identity mappings or 1×1 convolutions, to ensure a smooth information flow. After completing feature extraction, global average pooling (GAP) reduced the output to a 1×1×512 feature vector, which was then passed to a fully connected layer for binary classification purposes, as shown in Fig. 5b. The ResNet18 design is particularly

well suited for polar remote sensing tasks. Its residual connections allow the network to be trained effectively even at greater depths, combating gradient degradation and yielding improved accuracy. Moreover, the use of GAP instead of dense layers helps maintain its performance in cases with smaller input sizes, such as 4×4, making it effective for the automated detection of icebergs from remote sensing imagery.









**Figure 5: Workflow of the iceberg detection framework and architectural overviews of three comparative models. (a) shows the SVM; (b) shows the layered structure of ResNet18; (c) presents the hierarchical stages of the IDAS-Transformer architecture. Patch Partitioning: Transforming the input image into a sequence. (d) Two successive Swin transformer blocks. W-MSA: Multi-head self-attention with regular window configurations. SW-MSA: Multi-head self-attention with shifted-window configurations. MLP: multi-layer perceptron. LN: Layer normalization.**

As shown in Fig. 5, the process began with Sentinel-1A SAR HV polarization data, which underwent preprocessing, including ellipsoid correction, linear stretching, and transformation. Following image segmentation, the dataset was fed into three classification models: an SVM, ResNet18, and the proposed IDAS-Transformer.

To determine the number of icebergs, this study employed the Connected Components function from OpenCV, which identifies and labels distinct regions in binary images on the basis of pixel connectivity. By applying the 8-connected rule, each contiguous iceberg region was assigned a unique label, enabling efficient counting and analysis processes.

### 2.2.3 Performance metrics

To evaluate the performance of the three algorithms, we compared their results via manual interpretation using multiple quantitative metrics. Their equations are as follows:

$$Precision = \frac{TP}{TP+FP} \tag{1}$$

$$Recall = \frac{TP}{TP+FN} \tag{2}$$

$$F1 = 2 * \frac{Precision*Recall}{Precision+Recall} \tag{3}$$

where a true positive (TP) and a false positive (FP) represent cases in which the model predicted the positive class as being positive and negative, respectively, and a true negative (TN) and a false negative (FN) represent cases in which the model predicted the negative class as being negative and positive, respectively.

Iceberg areas were defined as the positive class, whereas the non-iceberg areas (i.e., the ocean) were defined as the negative class. The F1 score falls between 0 and 1. The closer its value is to 1, the better the performance of the corresponding model. In addition, the kappa coefficient is commonly used as a spatial consistency test for image classification tasks. It serves as a measure of classification accuracy. The associated calculation formula is as follows:

$$kappa = \frac{p_o - p_e}{1 - p_e} \tag{4}$$

$$p_o = \frac{TP+TN}{TP+TN+FP+FN} \tag{5}$$

$$p_e = \frac{(TP+FN)(TP+FP)+(TN+FN)(TN+FP)}{(TP+FP+TN+FN)^2} \tag{6}$$

where $p_o$ is the accuracy. It returns the rate of correct predictions without differentiating positives from negatives. p_e is the level of agreement expected by chance.



The count and area biases were used as quantitative metrics to assess the discrepancies between the model classification results
and manual interpretations. Specifically, each bias was calculated as the relative difference between the model-derived value
and the reference value obtained through manual interpretation, and then it was divided by the reference value and expressed
as a percentage. A positive bias indicated overestimation by the model, whereas a negative bias indicated underestimation.

## 3 Results

### 3.1 Iceberg identification (different sea ice backgrounds)

In AOI1, which is located in the landfast ice region of Prydz Bay, the SAR image presented an average sea ice concentration
of 75%. Here, the IDAS-Transformer identified 594 icebergs, slightly more than the 582 icebergs detected through visual
interpretation, resulting in a modest count bias of +2.1%. The total iceberg-covered area accounted for 1.92% of the SAR
image according to the IDAS-Transformer, closely matching the value of 1.82% derived from manual delineation. The area
bias was approximately +5.0%.

In AOI2, despite the presence of a similar sea ice concentration (73%) in a floe ice environment, the count bias increased
significantly to +11.2%, indicating a more pronounced discrepancy. The IDAS-Transformer detected 109 icebergs, whereas
visual interpretation identified 98 icebergs. However, both methods yielded nearly identical iceberg area fractions of
approximately 0.18%. The corresponding F1 score (86.46%) and kappa coefficient (86.44%) indicated solid performance,
albeit slightly lower than that achieved for AOI1 (Table 2).

When the SIC was reduced to 21% in the low-concentration floe ice region of the Ross Sea (AOI3), the IDAS-Transformer
identified 115 icebergs, whereas 105 icebergs were identified via visual interpretation, resulting in count and area biases of
+9.5% and −6.7%, respectively. The F1 score (85.52%) and kappa coefficient (85.50%) were slightly lower, indicating that
even with less sea ice, interference persisted and affected the detection accuracy of the model.

In AOI4, which is characterized by open water, icebergs could be easily and accurately detected from SAR images using both
automated and manual detection methods, with a total of 14 icebergs identified. Both approaches consistently identified 14
icebergs, but a slight difference in area was observed, possibly due to the blurred boundaries of ice floes.

The IDAS-Transformer tended to detect more icebergs than visual interpretation did, with a mean count bias of +4.13% and a
mean area bias of +3.65% across all regions. Figure 6 displays the results extracted from four AOIs under different ice
concentration conditions through both the IDAS-Transformer and manual interpretation. While the detection performance
remained stable, sea ice—particularly in floe ice environments—continued to exert a significant influence on the detection
outcomes.



**Table 2. The detailed metrics of the iceberg detection results.**

| Research area | | AOI1 | AOI2 | AOI3 | AOI4 |
|---|---|---|---|---|---|
| Number of icebergs | IDAS-Transformer | 594 | 109 | 115 | 14 |
| | Visual interpretation | 582 | 98 | 105 | 14 |
| Total iceberg area | IDAS-Transformer | 1.92% | 0.18% | 0.14% | $3.65\times10^{-4}$ |
| | Visual interpretation | 1.83% | 0.18% | 0.15% | $3.20\times10^{-4}$ |
| Count bias (%) | | +2.1 | +11.2 | +9.5 | 0 |
| Precision (%) | | 85.62 | 86.74 | 89.25 | 85.59 |
| Recall (%) | | 90.01 | 86.19 | 82.08 | 96.89 |
| F1 (%) | | 87.76 | 86.46 | 85.52 | 90.88 |
| Kappa coefficient (%) | | 87.53 | 86.44 | 85.50 | 90.87 |





**Figure 6: Iceberg identification results obtained under varying SIC conditions using the IDAS-Transformer framework. Panels a, d, g, and j display preprocessed Sentinel-1 SAR images. Panels b, e, h, and k show the ground-truth annotations derived from manual visual interpretations, and panels c, f, i, and l represent the automated iceberg detection results obtained from the IDAS-Transformer. (a–c) AOI1: landfast ice (SIC = 75%). (d–f) AOI2: high-concentration floe ice (SIC = 73%). (g–i) AOI3: low-concentration floe ice (SIC = 21%). (j–l) AOI4: ice-free area (SIC = 0%). Blue: icebergs; orange: other terrain features.**



## 3.2 Iceberg concentration

To support practical ship route planning and warning applications, the concept of iceberg concentration is introduced in this study. The iceberg concentration ($C_{iceberg}$ in Equation (7)) represents the proportion of the total iceberg area to the specified

grid cell area.

$$C_{iceberg} = \frac{A_{iceberg}}{A_{grid}} \tag{7}$$

where $A_{iceberg}$ is the area of identified icebergs within a grid cell and $A_{grid}$ refers to the area of the grid cell, which is defined on the basis of the chosen spatial resolution. The grid size can be flexibly defined depending on the specific requirements of the given application.

To refine the iceberg risk assessment process, we tested grid sizes of 10 km × 10 km and 1 km × 1 km. In AOI1, with a grid resolution of 10 km × 10 km, the maximum iceberg count within a single cell was 70, while the highest iceberg concentration reached 17.4%. The maximum average iceberg size was 2.5 km². However, for practical risk assessment purposes, such a coarse grid proved insufficient. To better capture the detailed variability, a finer resolution of 1 km × 1 km was adopted for subsequent analyses.

Figure 6a highlights a large iceberg near 67°35′S, 71°00′E in AOI1. At a 1 km × 1 km resolution, this iceberg spanned multiple grid cells. This resulted in a 100% iceberg concentration within these grid cells, although only one iceberg was present in total. In contrast, AOI2 (Fig. 7e) revealed a more even spatial distribution: most grid cells contained a single iceberg, occasionally two. AOI3 (Fig. 7h) followed a similar pattern. The maximum concentration in AOI2 reached 80%, indicating that icebergs covered 0.8 km² of the total 1-km² area of a single grid unit. In AOI3, the peak decreased to 30%, suggesting smaller average

iceberg sizes. As depicted in Fig. 6, in AOI2 and AOI3, the iceberg boundaries appeared visually indistinct, making direct interpretation difficult. The concept of iceberg concentration offers an effective way to describe both the number and sizes of icebergs within a grid, providing a clearer, more quantitative basis for evaluating navigational risk. This spatial metric enables a more intuitive decision-making process to be implemented in iceberg avoidance scenarios for polar shipping.

The results derived from AOI4 (Figs. 7j–l), an ice-free region, revealed a minimal iceberg presence. Most grid cells were

empty, confirming the low-risk environment.





**Figure 7: Spatial distributions of iceberg metrics across AOI1–4. Panels a, d, g, and j show the iceberg concentration, defined as the proportion of the grid area occupied by icebergs, in AOI1, AOI2, AOI3, and AOI4, respectively. Panels b, e, h, and k present the number of icebergs per 1 km × 1 km grid cell in each area. Panels c, f, i, and l display the average individual iceberg area within each grid cell, calculated as the total iceberg area divided by the number of icebergs. All results are based on a grid resolution of 1 km × 1 km.**




### 3.3 Application cases

The coastal area around Zhongshan Station in Prydz Bay, East Antarctica, is covered by landfast sea ice for most of the year. Annually, approximately 10–20 icebergs ground offshore from Zhongshan Station, with the nearest iceberg located within 1

km of the station. Some grounded icebergs persist in the region for several years, acting as barriers that suppress the dynamic disintegration of offshore landfast ice during the late summer months. In contrast, smaller icebergs typically drift away during the summer months. Despite the importance of precise spatiotemporal data for ensuring safe navigation in this complex marine environment, research on artificial intelligence-based iceberg detection methods for this region is still scarce.

We analysed SAR images acquired along the coast of Zhongshan Station, Antarctica, on 22 November 2023 using the proposed

iceberg identification method. The distribution of these icebergs is shown in Fig. 8a. The enlarged view (Fig. 8b) clearly indicates that the iceberg concentrations were notably higher in the nearshore areas near the station. Figure 8c shows the statistical distribution of the iceberg concentrations in 1 km × 1 km grid units, revealing peak concentrations of 70–80% in the western offshore area adjacent to the station. Figure 8d shows the iceberg counts per grid cell, with a maximum of 5 icebergs observed in a single cell; notably, most grid cells containing more than two icebergs were clustered close to the station. Finally,

Fig. 8e presents the average size of the individual icebergs within each 1 km × 1 km grid cell, with the majority of the icebergs near Zhongshan Station being smaller than 0.3 km².

Our method was successfully implemented in the coastal region of Qinling Station, demonstrating its utility in the Ross Sea coastal environment. Although the iceberg concentration in this region was not high enough to warrant a detailed representation here, the accelerating disintegration of ice shelves continues to generate icebergs. These icebergs exhibit dynamic mobility

through wind and ocean current forcings. Consequently, regardless of the abundance of icebergs, the potential for encounters remains a constant consideration for researchers studying ships in Antarctica.





**Figure 8: Iceberg characteristics derived from Sentinel-1 SAR imagery in Prydz Bay, East Antarctica (22 November 2023). (a) Spatial distribution of the icebergs located along the Prydz Bay coastline; (b) Enlarged view of the SAR-based iceberg detection results obtained near Zhongshan Station; (c) Iceberg concentrations (%) within 1 km × 1 km grid cells; (d) Number of icebergs per grid cell; (e) Average individual iceberg areas (km²) within the grid cells. The white dashed lines indicate the front of the Amery Ice Shelf after moving towards Prydz Bay and the landfast ice edge.**

## 3.4 Evaluations of three models

To comprehensively evaluate the overall performance of the proposed approach, we compared the IDAS-Transformer with two widely used methods: an SVM and the ResNet18 architecture. Figure 9 displays the comparative results obtained for AOI1, where 582 icebergs were identified visually. The detection counts for the SVM, ResNet18, and the proposed method were 631, 664, and 594, respectively, corresponding to calculated biases of +8.4%, +14.1%, and +2.1%, respectively. The comparative statistics produced for the remaining AOIs are summarized in Table 3. Among the three tested methods, the IDAS-Transformer



consistently achieved the best performance, with the lowest mean bias of +4.13%. The proposed model demonstrated robust

performance across all the study regions, with its F1 scores and kappa coefficients consistently exceeding 85%, outperforming

both baseline models by significant margins (Table 3). In contrast, the SVM showed considerable overestimation, with a mean

bias of +26.28%, and generally lower F1 and kappa coefficients, hovering at approximately 72% in AOI3. Especially in AOI2,

many instances where the SVM misclassified the edges of floe ice as icebergs were observed, largely accounting for its

significant overestimation effect.





**Figure 9: Comparison among the iceberg detection results produced by the three tested models for AOI1 (d-f). (a) MODIS image of AOI1 (30 August 2023); image credit: NASA Worldview (https://worldview.earthdata.nasa.gov/); (b) Sentinel-1 SAR input image; (c) Ground-truth annotations derived from manual visual interpretation; (d) Detection results obtained from SVM model; (e) Detection results obtained from ResNet18 model; (f) Detection results obtained from IDAS-Transformer model.**



**Table 3. F1 scores and kappa coefficients (%) yielded by the three models for the four AOIs.**

| AOI | F1 | | | Kappa coefficients | | |
|---|---|---|---|---|---|---|
| | SVM | ResNet18 | IDAS-Transformer | SVM | ResNet18 | IDAS-Transformer |
| 1 | 86.07 | 84.23 | 87.76 | 85.82 | 83.91 | 87.53 |
| 2 | 80.50 | 82.84 | 86.46 | 80.46 | 82.81 | 86.44 |
| 3 | 72.35 | 83.80 | 85.52 | 72.31 | 83.78 | 85.50 |
| 4 | 89.19 | 83.08 | 90.88 | 89.19 | 83.07 | 90.87 |

## 3.5 Iceberg size comparison

The three models exhibited notable differences in terms of detecting icebergs of varying sizes under diverse sea ice backgrounds, particularly for small targets. For icebergs smaller than 0.05 km², the SVM identified 576 objects as icebergs, resulting in a +128% overestimation rate relative to the 253 icebergs detected by manual interpretation. ResNet18 also exhibited a substantial bias of +50.6% (381 detections), whereas the IDAS-Transformer performed best, identifying 312 icebergs with a bias of +23.3%.

Although manual delineation may have introduced a conservative bias, the significant overdetection demonstrated by the SVM points to deeper issues in its small iceberg identification process under complex sea ice conditions. Here, mixed pixels and unclear boundaries blurred the distinction between icebergs and surrounding floe ice. Notably, this excessive number of false positives was not evenly distributed but was largely concentrated among icebergs smaller than 0.05 km², a size range where the limitations of this model became most apparent (Fig. 10). In AOI2, the region with a high sea ice concentration, the SVM frequently misclassified floe ice edges as icebergs, resulting in a pronounced overestimation rate. This suggests that the SVM, with its reliance on local pixel-level features, combined with the use of linear kernel functions, has a limited ability to perform effectively in scenes with high background noise. In such cases, subtle spectral or textural cues can easily lead to misclassification. Misclassifications at the pixel level are prone to amplification, with fragmented or blurred regions misinterpreted as distinct icebergs; as a result, an excessive number of false positives was produced. ResNet18, while more stable due to its convolutional architecture, suffered from resolution losses in the small object detection task due to the use of pooling operations, often failing to preserve the detailed contours of small icebergs and inflating their apparent sizes (Figs 11a, c, and f). It also tended to overfit bright regions, which further distorted the area estimations. In contrast, the IDAS-Transformer used attention mechanisms to selectively focus on informative regions, enhancing its ability to distinguish subtle differences between small icebergs and the surrounding sea ice. Its multiscale feature fusion process and contextual awareness design enable it to preserve spatial resolutions while also capturing more refined semantic features, enhancing its precision and stability, especially in challenging or unclear scenarios.

For the medium icebergs (0.05–0.5 km²), the three methods identified similar numbers of icebergs: 388 for the SVM, 497 for ResNet18, and 472 for the IDAS-Transformer. Compared with the 506 icebergs identified via visual interpretation, the average



biases of the models were −23.32%, −1.78%, and −6.72%. The SVM failed to effectively detect icebergs in the 0.05–0.50 km²
range. In comparison, ResNet18 produced an iceberg count that closely matched the manual delineation results, especially for
AOI1. However, its performance in terms of accuracy and consistency, as reflected by the F1 score and kappa coefficient
(Table 3), was inferior to that of the IDAS-Transformer. This finding indicates that despite the strong detection ability of
ResNet18, it tended to misclassify non-iceberg features as icebergs. On the other hand, the IDAS-Transformer not only
maintained a reasonable detection count but also demonstrated superior precision and robustness. These results highlight the
greater reliability of the IDAS-Transformer for remote sensing applications in polar regions, where operational consistency is
critical.

For icebergs larger than 0.5 km², the SVM identified 45 icebergs, ResNet18 identified 59 icebergs, and IDAS-Transformer
detected 48 icebergs, whereas visual interpretation revealed 40 icebergs. The overestimation exhibited by IDAS-Transformer
occurred only in AOI1, where edge adhesion cases (such as those located at 67°27′S, 70°00′E in Figs. 6a–c) led to the
misclassification of 2–3 fragmented small icebergs as a single large iceberg. This is the main reason why IDAS-Transformer
overestimated the number of icebergs larger than 0.5 km² and explains why, despite discrepancies in the iceberg counts, the
total iceberg areas determined for AOI1 were quite similar with both the IDAS-Transformer and visual interpretation (Table
425 2).

In AOI2, the SVM identified only one iceberg larger than 0.5 km² (Fig. 11a), whereas both visual interpretation and our method
identified two icebergs. This discrepancy arises from Fig. 11b, where the SVM failed to classify a disintegrating iceberg with
serrated edges and visible cracks (area: 0.496 km²). This iceberg, on the brink of breaking apart, was correctly identified by
the other methods as exceeding 0.5 km². A rough iceberg surface, which is heavily jagged with cracks, is a clear sign of
impending disintegration, whereas a smooth iceberg with no visible cracks is categorized as stable. We also observed that
ResNet18 struggled most with resolving fine textures, as shown in Fig. 11b.




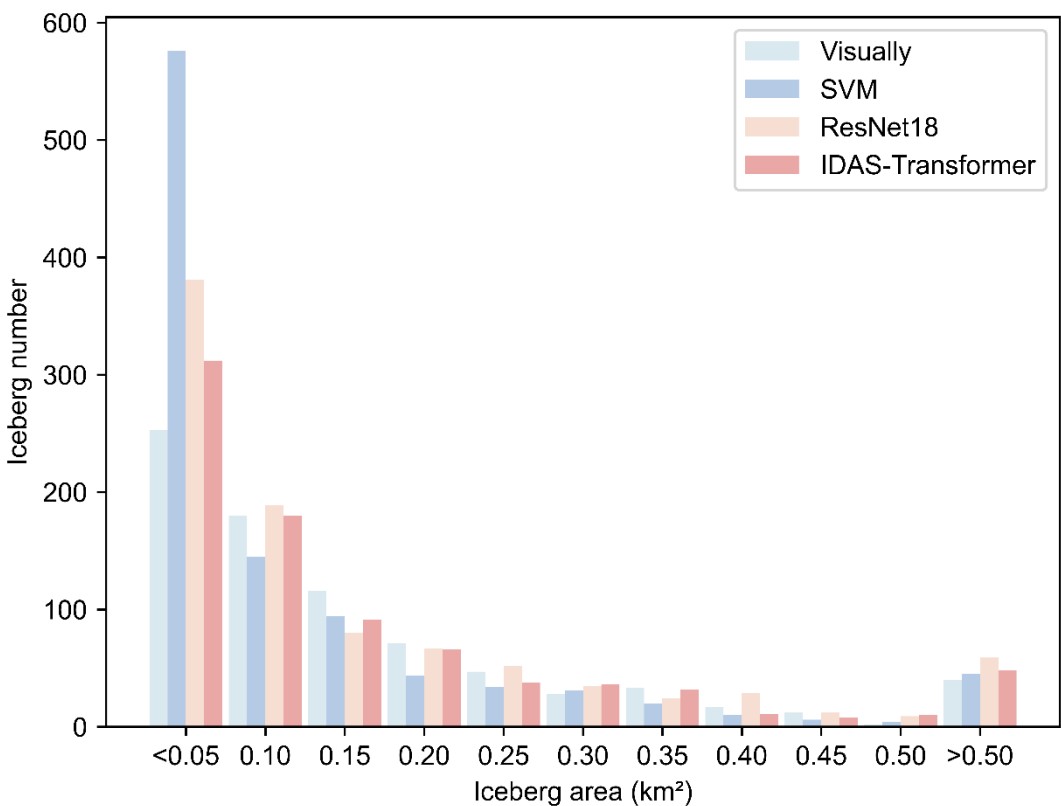

**Figure 10: The numbers of icebergs with various areas extracted from the three models and visual interpretation.**





**Figure 11: Performance comparison among three classification models across different iceberg size categories. Examples of input images (first column) and the results produced by manual interpretation (second column), our method (third column), the SVM (fourth column) and ResNet18 (last column). We picked these images to illustrate the detailed iceberg identification results. The numbers indicate the estimated iceberg areas (km²) yielded by each method.**



## 4 Discussions

In this study, we systematically investigated the iceberg detection task in the Prydz Bay and Ross Sea regions, addressing the current small-scale iceberg monitoring gap in these areas. Compared with the existing approaches, our method has clear advantages under complex sea ice conditions, particularly in terms of reducing the interference caused by fragmented ice edges. Early studies relied primarily on traditional thresholding or edge detection techniques. Young et al. (1998) detected icebergs in the seasonal sea ice zone of East Antarctica using adjusted thresholds, with a minimum detectable area of 0.06 km². However,

their results significantly underestimated the number of small icebergs and overestimated the average area by approximately 20%. Similarly, Wesche and Dierking (2012) reported both negative and positive deviations, but on average, the iceberg sizes were overestimated by $10 \pm 21\%$. Our IDAS-Transformer model also slightly overestimated the number of small icebergs, with the average area deviation controlled within +3.65%, indicating a notable improvement over these earlier methods.

In recent years, the adoption of advanced methods for detecting icebergs has increased; however, most of the focus remains

concentrated on the Amundsen and Weddell Seas and, predominantly, on large-scale icebergs. Mazur et al. (2017) developed an object-based segmentation approach using ENVISAT ASAR Wide Swath Mode (WSM)imagery to detect icebergs in the Amundsen Sea, generating 3.8% missing and 7.0% false-alarm rates. However, their detection accuracy markedly declined in spring and autumn, during wind-roughened water and in the presence of sea ice conditions, where the kappa coefficients decreased to 56.9% or lower. Notably, the highest number of misses occurred in iceberg clusters; this was likely due to the

presence of closely packed or frozen icebergs—a phenomenon that we also observed in AOI1 in our study (Figs. 6a–c), where adjacent icebergs within one-pixel proximity were rarely separated. Additionally, large icebergs lacking homogeneous textures occasionally resulted in undersegmentation in both our study and theirs (Fig. 11b). Despite these challenges, our model consistently maintained kappa coefficients above 85% across varying ice conditions, outperforming the previously developed approaches.

Koo et al. (2023) employed the Google Earth Engine to segment medium-to-large icebergs (0.4–10 km²) via simple noniterative clustering (SNIC) and a region adjacency graph (RAG), followed by SVM classification, achieving 99% accuracy. However, their method was tailored to localized regions and larger targets. Similarly, Braakmann-Folgmann et al. (2023) introduced a U-Net architecture for delineating giant icebergs in Sentinel-1 imagery, with an F1 score of 84%. However, the F1 score of the model decreased considerably in cases with coasts and dark icebergs.

A further distinction lies in the SAR data utilization and preprocessing pipelines of different methods. Most existing studies relied mainly on HH-polarized data (Mazur et al., 2017; Braakmann-Folgmann et al., 2023; Koo et al., 2023) in combination with different filters (Mazur et al., 2017; Barbat et al., 2021). These factors, as stated by Wesche and Dierking (2012), are sensitive to sea ice concentrations, atmospheric conditions, and seasonal variations. In contrast, our method adopts HV-polarized SAR imagery, utilizing ellipsoid correction, linear stretching, and linear transformation for preprocessing. This

simplification not only reduces the imposed computational burden but also validates the applicability of HV polarization for detecting icebergs in polar regions. Moreover, our dataset spans SAR images acquired in March, September, and December,



unlike in studies limited to winter images (Young et al., 1998), further underscoring the adaptability of the proposed model across different temporal conditions.

Certain methods, such as superpixel segmentation combined with ensemble learning (e.g., random forests), have achieved high detection accuracy (>97%) in specific regions, such as the Weddell Sea (Barbat et al., 2019a). However, these approaches typically require extensive training datasets and are based on various SAR sensor types (e.g., ERS, ENVISAT, Sentinel, and RADARSAT) and acquisition parameters, including spatial resolutions, polarization levels, and incidence angles. Each method has been applied to different regions, making it susceptible to regional variations and specific characteristics. In contrast, the transformer-based architecture of our model effectively captures both local features and the global context, demonstrating strong generalizability, particularly in complex sea ice conditions.

While the IDAS-Transformer model delivers marked improvements in small iceberg detection tasks and exhibits adaptability to complex ice conditions, certain challenges remain. Specifically, misclassification persists in regions with clustered icebergs or closely adjacent boundaries. Distinguishing between small icebergs and ice floes remains a significant challenge. Human expertise is also limited, making it difficult to accurately differentiate between small icebergs and ice floes, which can introduce errors in visual interpretation tasks. Additionally, discrepancies in the areas identified by the model may lead to inaccuracies in the subsequent statistical analyses of iceberg concentrations and simulations of iceberg meltwater. Future work may focus on further refining the model or integrating complementary techniques to improve its detection accuracy for small icebergs.

Satellite imaging and processing may inevitably result in temporal gaps (up to 1–2 days). These gaps may generate iceberg location displacements, leading to erroneous results, especially for freely drifting icebergs in the open ocean. In the packed ice zone, the location error of the iceberg displacement was small (Lichey et al., 2001). Nevertheless, incorporating satellite-based iceberg detection with an iceberg drift model would reduce such errors.

Notably, the IDAS-Transformer model system was deployed in a demo operation during the 40th Chinese Antarctic expedition in late 2023. The system identified iceberg locations along the cruise track to ensure navigation safety. Additionally, the IDAS-Transformer model system was deployed on the Tianhui cargo vessel (COSCO SHIPPING), which transported construction materials to establish China's new Antarctic coastal station (Qinling) in the Ross Sea. By processing near-real-time SAR imagery of the planned routes, the model generated daily iceberg distribution maps, providing approximately 60 updates during the voyage. These timely outputs enabled dynamic route optimization to be performed in response to evolving iceberg conditions, ensuring both the safety and efficiency of the mission. The success of this application demonstrates the robustness of the model under challenging Antarctic conditions and highlights its potential for broader integration into polar maritime safety frameworks.

## 5 Conclusions

The IDAS-Transformer, an iceberg detection algorithm based on the Swin transformer model, was developed. This is the first application of the Swin transformer architecture for iceberg detection. We demonstrated the robustness of the Swin transformer



in iceberg detection tasks. The IDAS-Transformer is capable of handling complex surface characteristic mixtures, including fast ice, pack ice, and open water, to identify icebergs. By incorporating AMSR2 SIC data, we successfully identified icebergs across four scenarios: landfast ice cover, high- and low-SIC coverage areas, and an ice-free open ocean area.

The IDAS-Transformer achieved F1 scores ranging from 85.52% to 90.88% and kappa coefficients ranging from 85.50% to 90.87% across the study areas in Prydz Bay and the Ross Sea. The IDAS-Transformer outperformed both an SVM and ResNet18 in terms of most metrics, achieving the lowest mean bias (+4.13%) alongside the highest F1 scores and kappa coefficients. In contrast, the SVM and ResNet18 exhibited considerably lower performance, with their F1 scores and kappa coefficients consistently below 90%. The mean biases of the iceberg count yielded by the SVM and ResNet18 were +26.28% and +17.27%, respectively, which were significantly greater than those of the IDAS-Transformer.

The results obtained from the IDAS-Transformer revealed optimal performance in the ice-free area, where all visually interpreted icebergs were successfully identified, and the model achieved F1 scores and kappa coefficients exceeding 90%.

In the landfast ice region, the bias of the IDAS-Transformer-identified iceberg count was +2.1%, whereas in the high- and low-SIC areas, the bias increased to +11.2% and +9.5%, respectively. These findings reveal that while SIC plays an important role, the primary challenge in iceberg detection tasks lies in distinguishing icebergs from surrounding pack ice.

We further investigated the differences among the models in terms of detecting small icebergs (<0.05 km²). The IDAS-Transformer maintained a relatively low count bias of +23.3% for small icebergs. In contrast, the SVM and ResNet18 exhibited higher count biases of +128% and +50.6%, respectively. Three primary factors likely account for these discrepancies: (1) challenges in distinguishing small icebergs from fragmented ice floes in complex zones, especially in the results of the SVM; (2) blurred iceberg boundaries, which may have led models to segment a single iceberg into multiple smaller icebergs; and (3) a conservative visual interpretation, which excluded ambiguous features.

IDAS-Transformer products can enhance the navigation safety of vessels operating in ice-covered waters and support the development of polar shipping routes. The iceberg concentration metric introduced in this study offers practical value for route planning tasks, marine operations, and iceberg hazard assessments.

The IDAS-Transformer possesses limitations with respect to detecting small icebergs, partly due to the inherent shortcomings of the manual SAR image interpretation process. This highlights the need for sufficient, onsite iceberg observations to validate remote sensing products (Karvonen et al., 2021).

The current ocean models typically omit grounded icebergs (e.g., Äijälä, et al., 2025). The incorporation of high-precision iceberg detection results can improve the model initialization effect and enhance the understanding of iceberg–sea ice–ocean interactions, leading to more accurate simulations. This integration scheme is particularly critical, as climate change is accelerating ice shelf disintegration, increasing the number of small icebergs. This in turn affects sea ice drift, but this process has not yet been incorporated into the existing models (Äijälä, et al., 2025).

The incorporation of DL-based iceberg detection algorithms with physics-based iceberg drift models can be the next step for developing an advanced real-time iceberg risk warning system.



*Code availability*. The code is available to authorized users on Zenodo at: https://doi.org/10.5281/zenodo.15600896, as specified in the access permissions.


*Data availability.* The Sentinel-1 SAR images are accessed from the Alaska Satellite Facility (available at https://search.asf.alaska.edu/). The MODIS images are obtained from https://worldview.earthdata.nasa.gov/. The AMSR2 data are sourced from https://data.seaice.uni-bremen.de/MultiYearIce/ascat-amsr2/raw/Antarctic/netcdf/. The processed datasets are openly available on Zenodo at: https://doi.org/10.5281/zenodo.15662494.


*Author contributions.* This study was devised by FM and JZ Initial data retrieval and analyses, as well as draft preparation, were performed by FM. BC contributed to draft preparation, results analysis, and editing. ZS and YC conducted modelling, data analyses, and contributed to draft manuscript preparation and editing. CJ, BC, KW, CW, YC, ZS, and JZ contributed to manuscript editing. All authors participated in the interpretation of the results and manuscript revisions.


*Competing interests.* Bin Cheng is a member of the editorial board of the journal "The Cryosphere".

*Acknowledgements*. This study was financially supported by the National Natural Science Foundation of China (grant nos. 42276251). This work was funded by the National Key Research and Development Program of China (grant no. 2022YFC2807003). Jiechen was supported by the Taishan Scholars Program. B.C. was supported by European Union's Horizon2020 research and Innovation Framework Programme PolarRES project [Grant No. 101003590]. Thank you especially to Lijian Shi for his financial support of the professional editing services, which greatly enhanced the grammar and language of this paper.

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
