# Peer review of "Iceberg Detection Based on the Swin Transformer Algorithm and SAR Imagery: Case Studies off Prydz Bay and the Ross Sea, Antarctic"

_EGUsphere, 2025_

## Author Comment (AC1)

The authors present a description of applying and adapting the Swin transformer architecture to the problem of detecting icebergs in satellite radar data. They compare their model performance against SVM and ResNet18 benchmarks and train and evaluate in a range of sea ice concentrations, with a view to developing a detection system that functions well across contexts. The contribution is interesting and a useful addition to the discussion, exploring how novel model architectures perform on a long-standing, important and as yet unsolved environmental monitoring problem. My opinion is that the manuscript contains some valuable contributions to the field but requires improvement before it is suitable to be published. My main concerns are around the lack of detail provided around methodological and design choices that have been made, and how these affect this study itself and the utility for the wider community. Some choices require stronger justification while others require more consideration and discussion of their implications. We also need to see full details of the machine learning training regimes to ensure that models have been appropriately and comparably trained, particularly considering the relatively small training dataset used. I have attempted to highlight where other detail would be beneficial in my specific comments below. Another general observation is that the literature cited is incomplete and somewhat outdated. I would recommend that the authors ensure they have read and included where appropriate all the relevant literature in this (still rather small) field. Figures are hard to interpret due to the scale. I would encourage the authors to show smaller areas in more detail where possible. Overall, I would consider that major revisions are likely to be required before publication, but the work definitely has the potential to form a valuable contribution to the field.

We sincerely thank you for your valuable comments and detailed suggestions. We also sincerely appreciate the reviewer's positive feedback. The revisions have substantially improved the clarity, completeness, and reproducibility of the manuscript, and have strengthened the justification of our methodological choices.

In the revised manuscript, we have made the following major changes:

**1. Methodological and design details**

Based on the referee's suggestion, we have added more motivation, justification and discussion of the methodological and design choices, including baseline selection, the choice of HV, replicating the same data three times, pre-processing. And we have provided a detailed description of the labelling process and the training configuration.

**2. Literature citation**

Based on the referee's suggestion, we have updated most of the pre-2010 references in the Introduction and added more related references to support our statements.

**3. Figures scale**

We fully understand your concern about the figures' scale. During manuscript preparation, we have been continuously adjusting the layout of the figures. To show more detail, we added each sub-panel as an individual high-resolution figure in the supplementary material of the updated manuscript. Please see below our response (text in red) and revised text (Italic text in red) to each of your comments (text in black) point-by-point.

**Specific points:**

Introduction: Many of these references look quite old and there is newer literature available to support most of these initial statements – Can the authors update their literature search, maybe

looking at authors like Coulon, Davison etc. to provide more recent relevant insights.

L68 – There are more studies out there. Consider citing

Evans et al. (2023): https://www.sciencedirect.com/science/article/pii/S0034425723003310

and Chen et al. (2025): https://essd.copernicus.org/preprints/essd-2025-51/

and Jafari et al. (2025) for the DL side: https://www.mdpi.com/2072-4292/17/4/702

Response: Thank you for your suggestion. We fully agree that including newer references can better reflect the current state of research. We also greatly appreciate your specific recommendations of relevant authors and papers. We carefully read these studies and found them highly informative and inspiring — in particular, Davison et al. (2020) highlight the importance of simulating submarine iceberg melting in studies that quantify interactions between the ice sheet and the ocean.

Following the reviewer's suggestion, we have made these changes in the revised manuscript.

We have updated most of the pre-2010 references in the Introduction to more recent studies.

The study of Coulon et al. (2024) has been cited on L43.

The three recommended references (Evans et al., 2023; Chen et al., 2025; Jafari et al., 2025) have been added on L68.

L85-89. These claims that SWIN transformer is better than other methods need to be substantiated by citations if they appear in this section. At present, they are not evidenced, and even if citations were provided, a little more quantitative detail on exactly how much better SWIN is than other methods for certain tasks would be needed. These statements seem to be the foundational premise for the choice of architecture in this study, but are currently unsupported by evidence, which raises questions over the chosen approach. Indeed, having claimed that SWIN transformer is better, on L95 the authors say it hasn't previously been applied to this question.

Response: Thank you for this valuable comment. In the revised manuscript, we have clarified and reorganized the foundational premise for the choice of architecture in this study (*L85-103*). Following the reviewer's suggestions, we have added supporting citations and more quantitative detail. We also reorganized the statement to ensure a more logical justification for selecting the Swin Transformer in this study. The revised text is provided below for your convenience. *L85-103*:

The transformers with global modelling capabilities have become new alternatives for visual tasks in satellite imagery (Dong et al., 2021; Huang et al., 2022; Wu et al., 2023; Shi et al., 2024), including applications in polar science (Sudakow et al., 2022; Ristea et al., 2023; Zhu et al., 2023). In Arctic studies, Sudakow et al. (2022) used a Swin Transformer architecture with a cross-channel attention decoder to detect the melt ponds on sea ice, which boosts the model's performance. Ristea et al. (2023) presented a novel approach outperforming classical convolutional networks for sea ice segmentation based on SAR imagery, which has a transformer core. In Antarctica, Zhu et al. (2023) developed a Swin transformer-based neural network (GLA-STDeepLab) for detecting ice shelf calving fronts, achieving results with an intersection over union (IoU) of 0.94 and a mean distance error (MDE) of  $473 \pm 34$  m. It is capable of identifying precise dynamic information about glaciers and ice shelf fronts. These studies demonstrated that the Swin transformer can capture both finegrained details and high-level contextual information while maintaining computational efficiency (Sudakow et al., 2022; Zhu et al., 2023). Its hierarchical structure empowers the model to handle diverse visual tasks, including object detection, image classification, and image segmentation (Meena et al., 2025). The calving front at an ocean boundary presents similar conditions and challenges to those encountered by icebergs (Braakmann-Folgmann et al., 2023). Because of the many successful studies using Swin transformer, including one addressing similar challenges (Zhu et al., 2023), we decided to employ Swin transformer to develop a novel iceberg detection algorithm called "IDAS-Transformer".

To our knowledge, it is the first study to apply the Swin transformer for iceberg detection tasks. We applied several architectural modifications to the Swin transformer to further accommodate the characteristics of Antarctic SAR imagery, with the aim of increasing its detection accuracy and noise resilience. These changes included tailored adjustments to mitigate speckle noise and optimizations of the depth and channel configurations of the model to improve its detection performance.

L115 – The authors select HV based on some evaluation of contrast. Please could they provide more detail on how this was conducted and present some of the data supporting their decision (possibly in supplementary material)? HV is not widely available across Antarctica compared to HH, so the choice to develop a system that may not be scalable because of the polarisations chosen probably needs a little more justification and some discussion of the trade-offs with generalisability that it implies.

Response: Thank you for this valuable comment. We appreciate the reviewer's suggestion to provide more detail regarding the choice of HV polarization. We have added a detailed explanation in *L122-131*, and supplementary material in *Appendix A* in the revised manuscript, including data and figures, to support our choice of HV polarization. We acknowledge that this design choice constrains generalizability. This study was conducted in two regions: Prydz Bay and the Ross Sea. We have added text in the Discussion to make this limitation explicit (*L518-521* in the revised manuscript). The revised text is provided below for your convenience.

L122-131:

Although the HH (horizontal transmit and horizontal receive) band is the primary polarization mode that is available in polar region, we evaluated both HH and HV bands to determine the suitable input data for iceberg detection. An analysis conducted during data preprocessing revealed that HV polarization offers better contrast and backscatter differences between iceberg and sea ice, especially for smaller icebergs, than does HH polarization (Fig. A1). At finer scales, the detailed surface features in Fig. 1 further support the choice of HV polarization in this study. This finding was validated through a cross-comparison conducted on 20 randomly selected SAR images, with over 90% of the results confirming the superior performance of HV. Similarly, Karvonen et al. (2021) observed that there is significantly less clutter and sea ice features visible in the HV than in HH polarization channel. Consequently, HV-polarized Sentinel-1 SAR imagery was chosen as the primary data source for this study. Appendix A1 provides more detailed information on how this analysis was conducted and presents the representative figures and histograms for supporting the choice.

L518-521:

HV provides a clearer separation between sea ice and icebergs. We acknowledge, however, that HV is not widely available across Antarctica compared to HH. We focus on two regions: Prydz Bay and

the Ross Sea in this study. To enhance the model's generalizability, future work will introduce HH polarisation and incorporate dual-polarization data.

Appendix A:

A1 Comparison of HH and HV polarization for iceberg detection

To select the most suitable polarization channel for iceberg detection, we compared the backscatter contrast between icebergs and surrounding features in HH and HV polarizations using imagery from AOII. For AOII, we applied orbit files, thermal noise removal, radiometric calibration to  $\sigma^0$  and conversion to dB. In Fig. A1, the pixel values are the radiometrically calibrated backscatter coefficient ( $\sigma^0$ ) expressed in dB. At the scene scale, HH polarization is sensitive to surface scattering, making it suitable for detecting icebergs in open water. However, sea ice also returns relatively strong HH backscatter, which reduces the contrast between sea ice and small icebergs and makes iceberg edges harder to separate from the surrounding ice. In contrast, HV polarization is more sensitive to volume scattering. Sea ice appears darker in HV, while icebergs remain bright and their boundaries are sharper.

The backscatter coefficient histograms (Fig. A2) illustrate this difference. HV shows a relatively narrow peak around -25 dB, whereas HH has a broader distribution centred near -15 dB. This means that HH provides useful contrast in open water, but in sea-ice conditions it offers weaker class separation than HV. Most previous iceberg-detection studies have relied on HH data. Our focus here is on scenes with substantial sea-ice cover, where confusion between sea ice and small icebergs is a key limitation.

Some studies only import HH band image to detect iceberg because the HH band is the primary mode that is available in polar regions (Braakmann-Folgmann et al., 2023; Koo et al., 2023; Chen et al., 2025). Koo et al. (2023) monitored icebergs in the Amundsen Sea. Chen et al. (2025) presents a comprehensive circum-Antarctic iceberg dataset. However, our study focuses on Prydz bay and the Ross Sea, where HV band imagery is totally available (Fig. A3). That is also one reason that we decide to use HV polarization.

At finer scales, the detailed surface features in Fig. 1 further support the choice of HV polarization in this study.

For these reasons, we adopt HV in this study. We note that dual-polarization data are likely to be more effective approach for future research.

Figure A1. Comparison of HV (left) and HH (right) band images (A011). The two Sentinel-1 images were processed through apply orbit files, thermal noise removal, radiometric calibration and

**converted to backscatter coefficients ( $\sigma^0$ ) in dB.**

Figure A2. Histograms of backscatter coefficients ( $\sigma^0$ , dB) for HV (a) and HH (b) polarizations from AOII

Figure A3. Sentinel-1A Mission Observation Scenario: Mode - Polarization- Geometry (https://sentiwiki.copernicus.eu/web/s1-mission#S1-Mission-Polarimetry)

Table 1 - I note that the AOIs represent different times of year. Backscatter and sea-ice contrast are known to vary seasonally, but this sampling is varying the seasonality at the same time as the sea ice concentration. Can the authors please expand on the motivation for these particular time points that do not allow for independent assessment of the effect of sea ice concentration on classifier

performance rather than controlling for seasonality while varying sea ice concentration? Such an experimental setup would imply introducing a greater latitudinal variation, which also creates complexities for any classifier. In general I would like to see more recognition and explanation of the implications of the design decisions being made for the performance within this study and the scope for generalising to other areas.

L453-454 – This discussion comes back to the effect of selecting AOIs that vary both sea ice concentration and seasonality. By my reading, the authors have not structured their sampling to directly evaluate the effect of seasonality on their classifier and should probably acknowledge this when drawing comparison with other studies. See also the claims made in L470-471 that don't feel to me to be robustly supported. In addition to Mazur et al. (2017), Evans et al. (2023) also explicitly evaluated the effect of seasonality on classifier performance, while the recent work of Chen et al. (2025) is an example of using October-only data.

Response: Thank you for your insightful comments. Both points relate to the same issue, so we address them together below.

We agree that this sampling is varying the seasonality at the same time as the sea ice concentration. Our goal was to ensure that testing data include scenes with diverse sea ice conditions and concentrations ranging from high to low, to enhance the model's robustness and generalization. Because the essence of our approach is image recognition, we consider that the model primarily relies on the relative contrast between iceberg and non-iceberg features, rather than on the physical value of  $\sigma^0$ . Therefore, we did not control intentionally for seasonality variation in this sampling process.

In addition to, thank you for your important suggestion. We recognize that this design couples seasonal and concentration variations, and we have clarified this sampling design in the revised manuscript (*L138-140*). We have also clarified the discussion part in the revised manuscript (*L522-528*) that we did not directly evaluate the effect of seasonality on our model and admit this when drawing comparison with other studies.

The revised text is provided below for your convenience.

**L138-140:**

The AOIs were selected to represent a gradient of sea ice conditions—from landfast ice (SIC = 75%) to high-concentration ice floes (SIC = 73%), low-concentration ice floes (SIC = 21%), and open water (SIC = 0%). This selection design is to capture diverse sea ice conditions and we did not control for seasonality.

**L522-528:**

In addition to Mazur et al. (2017), Evans et al. (2023) explicitly evaluated the effect of seasonality on classifier performance. Their results showed that classifier performance varied seasonally, being best in Winter and Spring and worst in Summer and Autumn (Evans et al., 2023). While some studies controlled for seasonality, Young et al. (1998) used only winter months images and the recent work of Chen et al. (2025) focused on October-only data. However, in our study, the sampling design did not directly evaluate the effect of seasonality on our model. In future work applying our method to an independent assessment of the effect of seasonality, more robust model will be developed to inform the estimation of seasonally varying uncertainties.

L149 - "Regions with high brightness levels in the SAR images were labelled icebergs." - it is

unclear at this stage how this was done, but this sentence implies it might have been thresholding, which is unlikely to be robust. On L153 the authors say that each patch was annotated. This implies a manual annotation. Can the authors explicitly describe the process? Was the annotator able to see the context of the patch? If so, how much context? At what resolution/zoom level were annotations decided upon? Were multiple human annotators used? Was any attempt made to assess annotator consistency?

As with much of the method description, the design choices need more explanation and justification. Why did the authors choose to annotate on the patch level rather than vectorising outlines? Why was 4\*4 the chosen patch size, and what are the implications and trade-offs of this compared to other patch sizes? What was done with patches that were only partially iceberg? Does the patch approach limit the ability of the transformer to learn to make precise boundary delineations? Response:

Thank you for your valuable comments on the data annotation and design selection. They helped us clarify the methodology more explicitly in the revised manuscript. The detailed annotation procedure and related information are provided below.

First, we agree that the original wording ("Regions with high brightness levels in the SAR images were labelled icebergs.") was misleading. Iceberg and non-iceberg annotations were conducted manually rather than through thresholding. Icebergs they usually present as relatively bright objects within C-band SAR imagery, particularly at high latitudes (Wesche and Dierking, 2012; Evans et al., 2023).

The annotation procedure was as follows (using Fig. 3 as an example). We first visually identified iceberg and non-iceberg areas in each full scene. For each iceberg (for example, the large iceberg in the lower-right corner of Fig. 3), we first cropped a minimum bounding rectangle and divided it into non-overlapping  $4 \times 4$  pixel patches. Patches that were clearly part of the iceberg were labelled as iceberg. After that, we manually filtered these patches and discarded any patch in which the iceberg covered less than 75% of the area. Similarly, rectangular areas were selected from the background regions and  $4\times4$  pixel patches were extracted to define the non-iceberg.

During annotation, the annotator always had access to the full-resolution SAR scene in Python while drawing the boxes. Decisions were therefore not made on isolated 4 × 4 tiles with no context. Instead, each patch was chosen based on its appearance in the full scene, including its surroundings, texture, and edges, and only then exported as a patch. All annotations in the dataset were produced by a single experienced annotator. We chose this approach to keep internal consistency.

We selected patch-based labels rather than manually vectorising outlines around every iceberg. This was mainly for efficiency and to avoid subjective boundary error. We acknowledge that a patch-based strategy can, in principle, reduce boundary precision. However, in practice the model still reproduces realistic iceberg outlines. As shown in Fig. 11, the predicted boundaries closely follow the visually interpreted edges, suggesting that patch-based labels have only a minor impact.

The choice of a  $4 \times 4$  pixel patch size (about  $160 \text{ m} \times 160 \text{ m}$  on the ground) reflects a trade-off. Smaller patches, such as  $2 \times 2$  pixels (about  $80 \text{ m} \times 80 \text{ m}$ ), tend to pick up bright sea ice and would likely increase false positives. Larger patches, such as  $8 \times 8$  pixels (about  $320 \text{ m} \times 320 \text{ m}$ ), would smooth over small icebergs and lower the effective detection resolution. The  $4 \times 4$  size balances these two effects. For patches that contained only part of an iceberg, we removed any patch where the iceberg covered less than 75% of the patch area.

Regarding whether patch-based prevents the model from learning sharp boundaries, patch labels are less precise than pixel-level polygons. However, the hierarchical self-attention in the Swin Transformer captures spatial features well, and the model is able to detect coherent iceberg edges even in complex boundary regions (Fig. 11). We have added these details to the revised manuscript.

L151-3: Why should replicating the same data three times result in better noise resilience and improved performance when it isn't providing any more information to the network? This may be another place where some supplementary material could be used to detail what tests were undertaken and elucidate and evidence why duplicating data helps. Alternatively, can the authors demonstrate from mathematical principles why the transformer architecture should perform better with three identical channels?

Response: Thank you for this important point. You are absolutely right that replicating the same data three times does not introduce any more information to the network. We have revised the manuscript to clarify this point (L170-172). This operation is needed to adapt the input to the standard Swin transformer architecture. Directly modifying the input layer to accommodate a single-channel input would require redefining the patch embedding and normalization layers, potentially compromising training stability and cross-study comparability. In contrast, duplicating the single-channel SAR data across three channels offers a simple and reproducible solution that maintains architectural integrity. This strategy has been widely adopted in SAR deep learning research (Yu et al., 2021; Zhang et al., 2021).

L157 – the training data were balanced for iceberg/non iceberg, but what was the balance for sea ice conditions? Again, contextual information like a histogram would help the reader interpret what the transformer is being shown.

Response: Thank you for this comment. We would like to clarify our design choice. We consider that the main goal in this study is to detect icebergs using a transformer model. Thus, maintaining a balance for iceberg/non iceberg in the training data is the first order of importance. By contrast, whether an iceberg appears in landfast ice, pack ice, or open water is not the main focus during model training and can be regarded as out of the scope of this study. Therefore, we prefer to retain our current design, which ensures balance between iceberg and non-iceberg samples, without imposing an additional balance among different background types. Moreover, the sea ice conditions, i.e., ice types, are the background information. Since our focus was on iceberg detection, we don't think it is necessary to separate labels for these specific background types, nor to enforce a fixed proportion among them.

L163 – Did the authors not apply orbit files, sensor calibrations, thermal noise removal or radiometric terrain calibration? These are standard procedures for generating robust, reusable data from Sentinel 1. If the authors chose not to follow standard procedures then their models will not be transferrable to other Level 1 or analysis-ready datasets. If they can demonstrate that excluding these steps produces better detection performance then there is an argument for using non-standard products, but this paragraph fails to adequately support that choice. Furthermore, some of the SNAP preprocessing would likely help to mitigate the cross-scene and cross-swath variability that the authors then apply some bespoke processing to overcome. Can the authors provide a more robust justification for their pre-processing choices and show that they translate into better model

**performance?**

Response: We thank the referee for the detailed question. Yes, we did not apply orbit files, sensor calibrations, thermal noise removal or radiometric calibration. We employed only ellipsoid correction, because the study area is predominantly oceanic and contains minimal topographic variation, making terrain correction unnecessary. And, we consider that our study focuses on image classification rather than quantitative retrieval. While sensor calibration and thermal noise removal are indeed important steps for quantitative retrieval, they are less critical for our image classification tasks. Our approach has been validated through accuracy assessments, showing reliable performance even without these preprocessing steps. This design also reduces computational cost and enables a more efficient workflow for rapid iceberg detection, which is advantageous for developing a NRT processing chain for iceberg detection.

**L170 – What is this linear stretching process and how was it conducted?**

Response: Thank you for raising this question. We acknowledge that the term "linear stretching" in the original manuscript was not sufficiently accurate. In the revised manuscript, we refer to this step as a square-root intensity compression. Specifically, following ellipsoid correction, we apply a fixed square-root transform to the SAR amplitude image (Fig. 4a to 4b). This nonlinear transform reduces saturation in very bright iceberg returns, and compresses the range of nearby sea ice and broken ice pixels. The result is that the iceberg is still the brightest object in the scene, but its outline is shown more clearly relative to the surrounding ice and water (Fig. 4b), which provides the model with a more consistent target shape to learn. This procedure is applied in the same way to all scenes, without any scene-specific manual adjustment.

L175 – If pre-processing is manually customised on a per scene or per-AOI basis, this introduces subjectivity is highly detrimental to the wider applicability of any trained models arising from this work. Can the authors justify their choice to use a highly manual pre-processing stage having argued in the introduction that they are testing the SWIN transformer to improve generalisability of detection methods and overcome issues like backscatter variability?

Response: Thank you for raising this valuable question. We fully understand the reviewer's concern about the potential subjectivity in scene preprocessing and its impact on the wider applicability. In our workflow, the preprocessing pipeline is mostly fixed. All images first undergo ellipsoid correction, followed by the same square-root intensity compression (Fig. 4a to 4b). These steps use

identical parameters for every scene and AOI and are not manually customised.

The only stage that involves manual adjustment is the final step (Fig. 4b to 4c), which is implemented in OpenCV. This step applied a simple linear scaling operation. In our case, the parameter value is adjusted within a small, predefined range to ensure consistent image visibility across scenes. The choice is not arbitrary but based on stable visual consistency and the imaging characteristics of the sensor. This ensures that the preprocessing remains systematic and reproducible rather than subjective.

We also agree that further automation would improve reproducibility. In future work, we plan to develop an automatic parameter selection module to standardize the preprocessing workflow.

Moreover, we acknowledge that the term "linear transformation" is too broad in this context, so we have replaced it with the more precise term "linear rescaling".

L219-220 – Again I feel that the design choice has not been fully justified. Are there examples of SVM used for iceberg detection on SAR (yes) that motivate its use as a benchmark here – or would picking a more recent/widely used approach from the iceberg detection literature as an initial benchmark allow for more meaningful evaluation of the SWIN transformer against the state-or-theart in the iceberg detection field specifically?

L230 – 235 – There is some duplication here of the justification for use of ResNet18. Again, why not benchmark on a DL architecture that has already been deployed for iceberg detection?

**Response to L219-220 and L230-235:**

Thank you for these valuable comments. Since these questions relate to the same issue, we address them together.

We totally agree that there is some duplication of the justification for use of SVM and ResNet18 as benchmarks. We appreciate the reviewer's suggestion to clarify the motivation for using SVM and ResNet18 as benchmarks. We have refined the logic for selecting baseline comparison methods to enhance clarity and coherence in the revised manuscript (*L244-258*).

We also understand your concern about why not benchmark on a DL architecture that has already been deployed for iceberg detection.

The main goal of this study was not simply to compare other existing iceberg detection models, but systematically evaluate three representative learning paradigms — traditional machine learning (SVM), convolutional neural networks (ResNet18), and transformer-based architectures (IDAS Transformer) — under the same SAR and sea-ice conditions.

Several DL architectures have already been deployed for iceberg detection, but these models differ greatly in their network structures, data requirements, and preprocessing steps, making direct benchmarking less comparable and reproducible.

In contrast, SVM and ResNet18 are well-established and widely used in SAR applications (Zhao et al., 2001; Shah Hosseini et al., 2011; Leigh et al., 2013; Karvonen et al., 2024; Asha et al., 2025). They are often adopted as baseline methods in SAR target detection (Yu et al., 2021), especially related to iceberg—ship discrimination (Bentes et al., 2016; Heiselberg et al., 2020; Yang et al., 2020). Their strengths and limitations have been known in many studies. SVMs, although robust for linearly separable features, tend to struggle with mixed scattering signals and speckle noise in heterogeneous sea-ice environments (Khaleghian et al., 2021; Adegun et al., 2023). CNNs, while effective at extracting local features, are limited by the convolutional locality and translational invariance of the CNN and cannot capture global semantic information (Adegun et al., 2023; Zhu et al., 2023; Wang et al., 2024), significantly restricting the accurate delineation of iceberg outlines from surrounding sea-ice structures (Heiselberg et al., 2022).

The proposed IDAS-Transformer, adapted from the Swin Transformer, was designed to overcome these challenges by combining hierarchical feature representation with a shifted-window self-attention mechanism. This allows the model to jointly learn local texture variations and global contextual cues, resulting in more robust iceberg detection across diverse sea-ice conditions.

By comparing these three representative paradigms, this study demonstrates that the transformer-based architecture (IDAS-Transformer) effectively addresses the well-known limitations of traditional machine learning and CNN-based approaches. Therefore, SVM and ResNet18 were chosen as reproducible and representative baselines to clearly show how the transformer-based

framework largely mitigates these issues.

L240-244 – These claims about the suitability of ResNet for polar and specifically iceberg tasks need references to support them.

Response: Thank you for your suggestion. We have adjusted these claims on *L244-245* in the revised manuscript and added related references to support them.

L251 – "following segmentation" – Are the authors referring to patching here? Segmentation is surely the task of the models?

Response: We sincerely thank the referee for the correction. Our use of the term "segmentation" was ambiguous and could be misinterpreted as the model's task. We have replaced "following segmentation" with a more precise description, "after the preprocessed image was divided into small patches of 4x4 pixels, these patches were fed into,,,".

General: Where can I see the config for the various training runs? What hypaerparameters were explored and how? How long was each model trained for? What was the stopping criterion? These need to be presented to convince the reader that the differences in performance arise from greater architectural suitability of the transformer rather than hyperparameter choices or training regime/effort.

Response: Thank you for pointing this out. We have added detailed information in the revised manuscript (L173-181).

All models in this study were trained and evaluated on a single NVIDIA GeForce RTX 3060 GPU (6 GB memory) using Python 3.8 and TensorFlow. In all experiments, the batch size was fixed at 32. Each model was trained for up to 200 epochs with an early-stopping patience of 50 epochs. We used the Adam optimizer with an initial learning rate of  $1 \times 10^{-3}$  and applied L2 regularization to mitigate overfitting. When the validation loss did not decrease for 25 consecutive epochs, the learning rate was reduced by half. Training was automatically terminated once the validation loss reached a stable minimum. The datasets were split into training and validation sets using an 8:2 ratio. The test set consisted of four entirely new Sentinel-1 SAR scenes, and performance was evaluated using precision, recall, F1 score, and kappa coefficient. To ensure comparability, all models (SVM, ResNet18, and the proposed IDAS-Transformer) were trained in the same computational environment and used identical data splits, batch sizes, optimizers, loss functions, and early-stopping criteria. No task-specific or model-specific hyperparameter tuning was applied.

L474 onwards – The authors claim that previous superpixel approaches require a lot of training data, but transformers are notoriously data-hungry yet have been selected here for a relatively small study. This leads me to a general observation about the proposed study – which is that it is reliant on a fairly small dataset for training (particularly in the context of transformer architectures), and is developed and tested on small AOIs. This is not necessarily problematic, but the reader needs to be convinced that the size of the model is appropriate to the relatively small training set and that claims of generalisability such as deployment on vessels are justified. The authors should therefore present more detail on how they have guarded against overfitting and overparameterisation during model selection and training. This could be in supplementary material but will be needed to convince the audience that the approach is appropriate to the scale of the data and robust.

Response: Thank you for this thoughtful and important comment. We have added detailed

information on the model selection and the training regimes against overfitting and overparameterisation in the revised manuscript.

In this work, we use a Swin Transformer architecture that is hierarchical. Its shifted-window attention greatly lowers the number of parameters and the computational cost compared with global self-attention architectures such as ViT, while still allowing the network to capture the long-range spatial structure needed to distinguish icebergs from surrounding sea ice. To further avoid over-parameterization, we narrowed the model width by reducing the channel dimensions across the four stages from 64-512 to 32-256. We also applied L2 regularization, trained with the Adam optimizer, and used early stopping. The learning rate was reduced by half if the validation loss did not improve for 25 consecutive epochs, and training was stopped once the loss had stabilized, which helped the model converge smoothly without overfitting. In addition to these controlled training settings, the final model has already been deployed for iceberg detection on board the Tianhui cargo vessel during China's 40th Antarctic Scientific Expedition, where it performed reliably under real operating conditions. These results indicate that the IDAS-Transformer configuration used in this study is appropriate for the available data volume and is able to deliver stable, generalizable performance across different environments.

Figure 6 – These panels are a bit too small to easily see how well the classifier is detecting smaller icebergs and delineated boundaries – can the authors provide a zoomed-in panel in this or another figure showing detail of the annotations and predictions please?

Figure 7 – It may be related to the scale of these images again but it is hard for the reader to appreciate what benefit converting to iceberg concentration provides over simply giving presence/absence. Is this more evident on a larger-scale figure? Do the authors have an example of a standard route for a research vessel to a station and how the navigator would find the concentration product more useful than the raw iceberg map when traversing it? Something like that would give the reader a better appreciation of how this may be useful in support of safer navigation. I see something akin to this in Fig. 8, but even at that scale it is hard to appreciate what value the gridded concentration or count products provide over the raw detections.

Figure 9 – Again, at this scale it is hard to see any differences between the three approaches – can the authors provide zoomed-in views to allow the reader to see the nuances of how the different methods classify icebergs?

Response (Fig. 6,7,9):

Since these comments are closely related, we respond to them together. We have been thinking carefully about the issue of figure scale and have already made several adjustments during revision. We will provide high resolution subfigures in the supplementary material of the revised manuscript, to ensure the figures can be zoomed in to see details.

Thank you for raising the point about navigation (Fig. 7). We appreciate the reviewer's interest in how the gridded iceberg concentration product could be used in practice. At the moment, we do not have access to a standard or routinely repeated ship route for Antarctic research vessels.

We completely agree that showing a concrete navigation example would make the usefulness of the gridded iceberg concentration and iceberg count maps much clearer. In the revised manuscript (L395-398), We have added a brief explanation of how these products could support vessel. In

future work, once suitable ship-track data are available, we plan to include a dedicated case study that combines real navigation routes with these maps to demonstrate their operational value for safe vessel access.

L395-398:

The gridded iceberg concentration and iceberg count products are not only useful for describing iceberg distribution, but could also support navigation in coastal waters. Near Zhongshan Station in Prydz Bay, Figure 8c shows where icebergs are most densely clustered and where lower concentration corridors exist, which could help vessels plan safer routes to the station. Although we do not yet include an example ship track, this is a direct example of how the product could be used in practice.

L313 – I disagree that the authors introduce the concept of iceberg concentration for the first time (as implied in my reading of this line). Please rephrase.

Response: Indeed, "introduced" sound over expressed. The new sentence reads: To support practical ship route planning and warning applications, the concept of iceberg concentration is applied in this study.

L349 – Was this an application of the SWIN classifier to a completely previously unseen image? In this paragraph it is a little unclear. If so, this is a potentially a good test of its generalisability, having been trained on the other four scenes and arguable could offer more insight into the value of the approach than the patch-wise validation carried out on the same scenes as the training. If so, I would encourage the authors to provide evaluation metrics for this example too, and elaborate more on what these say about the method's wider applicability.

Response: We thank the referee for the positive feedback. Yes, this was an application of the model to a completely previously unseen image. We appreciate the reviewer's suggestion to provide evaluation metrics for this example, and we agree that this would strengthen the manuscript. Due to time constraints, we plan to add this analysis after the response period.

Section 3.4: I was expecting to see this earlier – to me it would make sense to present this alongside the transformer evaluation, and before the use-case example. Please consider moving this results section up the manuscript to sit more comfortably alongside the other performance metric descriptions, although the evaluation/discussion should remain in the discussion section of the manuscript.

Response: Thank you for pointing out this. The other reviewer also commented on this issue. Based on the suggestion from both reviewers, we moved sections 3.4 and 3.5 prior to section 3.3 Application cases. The order of figures is updated accordingly.

Thank you for the explanation and detail on the failure modes – this provides useful context to those thinking about building on this work. In L421-424 the conflation of smaller touching icebergs into a single predicted object is argued to result in a positive prediction bias in terms of object count – surely this is the other way around and should reduce the number of predicted objects compared to manual delineation?

Response: We thank the referee for the positive feedback and appreciate the careful correction. We have rechecked both the data and the interpretation. The numbers reported in this section refer

specifically to icebergs larger than 0.5 km². In this size range, the merging of several nearby small icebergs into a single object does not reduce the count; instead, it can produce objects whose total area exceeds 0.5 km². As a result, the model can predict more icebergs larger than 0.5 km² than that from manual interpretation. Therefore, the data and the reasoning in this part of the manuscript are consistent.

L429-430: "A rough iceberg surface, which is heavily jagged with cracks, is a clear sign of impending disintegration, whereas a smooth iceberg with no visible cracks is categorized as stable"

- what is the evidence for this and can the authors either provide references or remove this assertion? Surface cracking and texture are a product of the stress history of the ice and may not imply imminent fragmentation – some large tabular icebergs have substantial surface texture and crevassing but remain stable for decades – e.g. B-22. If the authors do have studies that link surface characteristics to a predictive capability regarding fracture I would be fascinated to read them and would encourage them to cite them here.

Response: Thank you for your insightful and valuable comment. Upon careful review of the literature, we could not find published references to support this statement. We apologize for this inaccuracy and have removed this assertion in the revised manuscript.

Figure 11: This is a really useful figure and should form the basis for discussion of the effect of classifying on a patch basis on predictions.

Response: Thank you for your positive feedback and suggestion. We have added some discussion of the effect of classifying on a patch basis on predictions in the revised manuscript (L473-477). L473-477:

We also observed that ResNet18 struggled most with resolving fine textures, as shown in Fig. 11b. By contrast, our model produced iceberg outlines that were broadly consistent with the manual delineations. The fact that the outline discrepancies are smaller than those of ResNet18, suggests that our Transformer architecture (IDAS-Transformer), trained on the patch level rather than vectorised outlines, can capture broader spatial context through its self-attention mechanism.

L488-490 – Can the authors explain why iceberg displacement is a problem for the specific task being addressed here of detection? Surely they are where they are when the image is acquired? I can see that it is an issue for providing data to mariners or other science questions. Can the authors clarify this statement to be explicit about why the temporally sparse satellite acquisitions might be challenging and for what purposes?

Response: We thank the reviewer for this helpful comment. We agree that iceberg displacement does not affect detection within a single image. It is an issue for providing data to mariners.

In the original manuscript, Lines 481–487 focused on model related sources of error. We realize that this may have led readers to interpret Lines 488–490 as implying that iceberg displacement is a problem for the detection task itself. We apologize for this ambiguity. We have revised the statement in this section to clarify the intended point about iceberg displacement and its impact. The updated text is provided below.

L543-547:

Satellite imaging and processing may inevitably result in temporal gaps (up to 1-2 days). These

gaps may generate iceberg location displacements, especially for freely drifting icebergs in the open ocean. In the packed ice zone, the iceberg displacement was small (Lichey et al., 2001). It is an issue for providing iceberg data to mariners. Furthermore, temporally sparse might be challenging about multi-temporal analyses and meltwater simulations. As a result, it is essential to incorporate satellite-based iceberg detection with an iceberg drift model.

L513 - This is also the setting where other methods (including CFAR) perform best, and where such methods may outperform this approach. Can the authors briefly contextualise their model's performance against the best available previous study in similar sea ice contexts for each of their settings? (This may not be for the conclusion, but would be nice to see somewhere).

Response: Thank for your comment. We agree that other methods perform best in open water and some results from other methods may outperform our approach. Following your suggestion, we have added discussion of model's performance against the best available previous study in open water in the revised manuscript.

**L542-549:**

Notably, other methods in the previous studies also performed very well in open ocean across both thresholding and machine learning, and may outperform our approach (Karvonen et al., 2021; Braakmann-Folgmann et al., 2023; Jafari et al., 2025). For example, Braakmann-Folgmann et al. (2023) applied U-net for mapping the extent of giant Antarctic icebergs, with Otsu and k-means baselines. All methods perform very well with F1 scores of 0.93–0.95 and the median absolute deviation (MAD) in area of 2.4 %–3.2 % in open ocean.

L530 – This discussion of modelling effects feels a bit 'out of the blue' – in that none of the previous discussion (except very early and briefly in the introduction) relates to model implementations. Does it need to be here? If so, should it be accompanied by a bit more detail about how this detection method contributes earlier in the discussion rather than appearing just in the conclusion?

Response: We want to emphasize that the iceberg is not only critical for marine operations but also an important component for ocean modelling. We admit the appearance of the text is a bit bumpy; therefore, we added one sentence to smooth the text flow. We place this paragraph here to highlight the impact and outlook of this work, and further applications of iceberg detection. We added the following sentence in the beginning of this paragraph: Iceberg is not only critical for marine operations but also an important component for ocean modelling. The current ocean models,,

**Minor points:**

L51 – carving à calving

Response: Done, corrected.

**References:**

Adegun, A. A., Viriri, S., and Tapamo, J. R.: Review of deep learning methods for remote sensing satellite images classification: experimental survey and comparative analysis, J. Big Dataa, 10, 93, doi:10.1186/s40537-023-00772-x, 2023.

Asha, N., Ananthakrishnan, G., Gokulakrishnan, K., & Sadhana, S.: Advancement in Satellite-Based Iceberg Detection: A Deep Learning Approach, In 2025 4th OPJU International Technology Conference

(OTCON) on Smart Computing for Innovation and Advancement in Industry 5.0, Raigarh, India, 09-11 April 2025, 1-6, 2025.

Bentes, C., Frost, A., Velotto, D., and Tings, B.: Ship-iceberg discrimination with convolutional neural networks in high-resolution SAR images, Proc. EUSAR 2016: 11th European Conference on Synthetic Aperture Radar, Hamburg, Germany, 06-09 June 2016, 1–4, VDE, ISBN 978-3-8007-4228-8 / ISSN 2197-4403, 2016.

Braakmann-Folgmann, A., Shepherd, A., Hogg, D., and Redmond, E.: Mapping the extent of giant Antarctic icebergs with deep learning, The Cryosphere, 17, 4675–4690, doi:10.5194/tc-17-4675-2023, 2023.

Chen, Z., Liu, X., Guan, Z., Li, T., Cheng, X., Li, T., Liu, Y., Liang, Q., Zheng, L., and Liu, J.: A Sixyear circum-Antarctic icebergs dataset (2018 – 2023), Earth Syst. Sci. Data Discuss. [preprint], doi:10.5194/essd-2025-51, in review, 2025.

Davison, B. J., Cowton, T., Sole, A., Cottier, F., and Nienow, P.: Modelling the effect of submarine iceberg melting on glacier-adjacent water properties, The Cryosphere, 16, 1181–1196, doi:10.5194/tc-16-1181-2022, 2022.

Dong, H., Zhang, L., and Zou, B.: Exploring vision transformers for polarimetric SAR image classification, IEEE Trans. Geosci. Remote Sens., 60: 1-15, doi:10.1109/TGRS.2021.3137383, 2021.

Evans, B., Faul, A., Fleming, A., Vaughan, D. G., and Hosking, J. S.: Unsupervised machine learning detection of iceberg populations within sea ice from dual-polarisation SAR imagery, Remote Sens. Environ., 297, 113780, doi:10.1016/j.rse.2023.113780, 2023.

Færch, L., Dierking, W., Hughes, N., and Doulgeris, A. P.: A comparison of constant false alarm rate object detection algorithms for iceberg identification in L- and C-band SAR imagery of the Labrador Sea, The Cryosphere, 17, 5335–5355, doi:10.5194/tc-17-5335-2023, 2023.

Heiselberg, H.: Ship-Iceberg Classification in SAR and Multispectral Satellite Images with Neural Networks, Remote Sens., 12, 2353, doi:10.3390/rs12152353, 2020.

Heiselberg, P., Sørensen, K. A., Heiselberg, H., and Andersen, O. B.: SAR Ship–Iceberg Discrimination in Arctic Conditions Using Deep Learning, Remote Sens., 14, 2236. doi:10.3390/rs14092236, 2022.

Huang, L., Wang, F., Zhang, Y., Xu, Q.: Fine-Grained Ship Classification by Combining CNN and Swin Transformer, Remote Sens., 14, 3087, doi:10.3390/rs14133087, 2022.

Jafari, Z., Bobby, P., Karami, E., and Taylor, R.: Machine Learning-Based Detection of Icebergs in Sea Ice and Open Water Using SAR Imagery, Remote Sens., 17, 702, doi:10.3390/rs17040702, 2025.

Karvonen, J., Gegiuc, A., Niskanen, T., Montonen, A., Buus-Hinkler, J., and Rinne, E.: Iceberg detection in dual-polarized C-band SAR imagery by segmentation and nonparametric CFAR (SnP-CFAR), IEEE Trans. Geosci. Remote Sens., 60, 1–12, doi:10.1109/TGRS.2021.3070312, 2021.

Karvonen, J.: U-net with ResNet-34 backbone for dual-polarized C-band Baltic Sea-ice SAR segmentation, Ann. Glaciol., 65, e32, doi:10.1017/aog.2024.33, 2024.

Khaleghian, S., Ullah, H., Kræmer, T., Hughes, N., Eltoft, T., and Marinoni, A.: Sea Ice Classification of SAR Imagery Based on Convolution Neural Networks, Remote Sens., 13, 1734, doi:10.3390/rs13091734, 2021.

Koo, Y., Xie, H., Mahmoud, H., Iqrah, J. M., and Ackley, S. F.: Automated detection and tracking of medium-large icebergs from Sentinel-1 imagery using Google Earth Engine, Remote Sens. Environ., 296, 113731, doi:10.1016/j.rse.2023.113731, 2023.

Leigh, S., Wang, Z., and Clausi, D. A.: Automated ice—water classification using dual polarization SAR satellite imagery, IEEE Trans. Geosci. Remote Sens., 52, 5529-5539, doi:10.1109/TGRS.2013.229023

1, 2013.

Meena, T., Vijaya, J., and Harsha, B.: Swin Transformers for Remote Sensing SAR Image Classificatio n, in:2025 IEEE International Conference on Emerging Technologies and Applications (MPSec ICET A), IEEE, Gwalior, India, 21-23 February 2025, 1-6, doi: 10.1109/MPSecICETA64837.2025.1111872 6, 2025.

Ristea, N. C., Anghel, A., and Datcu, M.: Sea ice segmentation from SAR data by convolutional transformer networks, in: IGARSS 2023-2023 IEEE International Geoscience and Remote Sensing Symposium, IEEE, Pasadena, CA, USA, 16-21 July 2023, 168-171, 2023.

Shah Hosseini, R., Entezari, I., Homayouni, S., Motagh, M., and Mansouri, B.: Classification of polari metric SAR images using Support Vector Machines, Can. J. of Remote Sens., 37, 220-233, doi:10.5589/m11-029, 2011.

Shi, C., Shen, D., Ma, X., and Wang, H.: Oriented Ship Detection in SAR Image with Data Augmentati on Based on Swin Transformer, in: 2024 39th Youth Academic Annual Conference of Chinese Associa tion of Automation (YAC), Dalian, China, 7-9 June 2024, 434-439, doi:10.1109/YAC63405.2024.1059 8538, 2024.

Sudakow, I., Asari, V. K., Liu, R., and Demchev, D.: MeltPondNet: A Swin transformer U-Net for dete ction of melt ponds on Arctic sea ice, IEEE J. Sel. Top. Appl. Earth Obs. Remote Sens., 15, 8776-8784, doi: 10.1109/JSTARS.2022.3213192, 2022.

Wang, R., Ma, L., He, G., Johnson, B. A., Yan, Z, Chang, M., and Liang, Y.: Transformers for Remote Sensing: A Systematic Review and Analysis, Sensors.24, 3495, doi:10.3390/s24113495, 2024.

Wu, H., Yu, L., Li, X., Zhou, L., Zhang, W., and Bai, G.: CTF-Net: a convolutional and transformer fus ion network for SAR ship detection, IEEE Geosci. Remote Sens. Lett., 20, 1–5, doi:10.1109/LGRS.202 3.3310206, 2023.

Yang, X., and Ding, J.: A computational framework for iceberg and ship discrimination: Case study on Kaggle competition, IEEE Access, 8, 82320-82327, doi: 10.1109/ACCESS.2020.2990985, 2020.

Yu, J., Zhou, G., Zhou, S., and Yin, J.: A lightweight fully convolutional neural network for SAR auto matic target recognition, Remote Sens., 13, 3029, doi:10.3390/rs13153029, 2021.

Zhang, T., Zhang, X., Li, J., Xu, X., Wang, B., Zhan, X., Xu, Y., Ke, X., Zeng, T., Su, H., Ahmad, I., Pan, D., Liu, C., Zhou, Y., Shi, J., and Wei, S.: SAR ship detection dataset (SSDD): Official release and comprehensive data analysis, Remote Sens., 13, 3690, doi:10.3390/rs13183690, 2021.

Zhao, Q., and Principe, J. C.: Support vector machines for SAR automatic target recognition, IEEE Trans. Aerosp. Electron. Syst., 37, 643-654, doi:10.1109/7.937475, 2002.

Zhu, Q., Guo, H., Zhang, L., Liang, D., Wu, Z., Liu, Y., and Lv, Z.: GLA-STDeepLab: SAR enhancing glacier and ice shelf front detection using swin-TransDeepLab with global-local attention, IEEE Trans. Geosci. Remote Sens., 62, 5218113, doi:10.1109/TGRS.2023.3324404, 2023.